# Hierarchical Quantized Diffusion Based Tree Generation Method for Hierarchical Representation and Lineage Analysis

## Abstract

In single-cell research, tracing and analyzing high-throughput single-cell differentiation trajectories is crucial for understanding complex biological processes. Key to this is the modeling and generation of hierarchical data that represents the intrinsic structure within datasets. Traditional methods face limitations in terms of computational cost, performance, generative capacity, and stability. Recent VAE-based approaches have made strides in addressing these challenges but still require branch-specific network modules for each tree branch, limiting their stability and scalability to deep hierarchies, while also suffering from posterior collapse. To overcome these challenges, we introduce HDTree, a diffusion-based approach that captures tree relationships within a hierarchical latent space using a unified hierarchical codebook and quantized diffusion processes to model tree node transitions. This method improves stability by eliminating branch-specific modules and enhances generative capacity through gradual hierarchical changes simulated by the diffusion process. HDTree's effectiveness is demonstrated through comparisons on both general-purpose and single-cell datasets, where it outperforms existing methods in reconstruction quality, generation diversity, and hierarchical consistency. These contributions provide a new tool for hierarchical lineage analysis, enabling more accurate and efficient modeling of cellular differentiation paths and offering insights for downstream biological tasks. (code: `https://anonymous.4open.science/r/code_HDTree_review-A8DB`).

## 1 Introduction

In single-cell research, tracing and analyzing cellular differentiation trajectories is essential for understanding dynamic biological processes. This task requires not only effective modeling of hierarchical structures (Zeng et al., 2022), but also the ability to generate data that faithfully captures such hierarchies (Guo et al., 2024). Accurately characterizing the hierarchical organization underlying cell differentiation facilitates the exploration of cellular systems and fate decisions, while conditional generation based on these hierarchies enables interpretable discovery of biological mechanisms. Importantly, hierarchical structures are not exclusive to biology—they also emerge in various domains such as recommendation systems, molecular design, and knowledge representation (Chehreghani & Chehreghani, 2024; Gyurek et al., 2024; Tian et al., 2024). Therefore, developing models that can both represent and generate data along hierarchical relationships not only enhances performance in downstream tasks such as classification and clustering, but also provides deeper insight into the intrinsic organization of complex data.

As shown in Fig. 1, Traditional methods (Murtagh & Legendre, 2014; Rokhlin & Tygert, 2017) often rely on a combination of dimension reduction (Jia et al., 2022), clustering (Oti & Olusola, 2024), and data regression (Ali & Younas, 2021) techniques to achieve hierarchical modeling and data generation. While these approaches can address the tasks to some extent, they face challenges regarding computation costs, performance, generative capacity, and stability (Zang et al., 2024b). These limitations make them inadequate for handling the demands of large-scale, high-dimensional biological data. Recent state-of-the-art (SOTA) methods based on Variational Autoencoders (VAEs) (Manduchi et al., 2023; Xiao & Su, 2024; Majima et al., 2024) have unified generative tasks and hierarchical representation within a single modeling framework, achieving notable advancements. These VAE-based

Figure 1: **The motivations.** Base method and deep method cannot meet the requirements of hierarchical representation and lineage analysis in terms of stability, generalivity, accuracy, and training cost.

approaches effectively reduce computational costs when processing large-scale data while improving performance and data generation capabilities. However, a key limitation of existing SOTA methods lies in their reliance on specialized network modules for each tree branch (Manduchi et al., 2023). This design not only reduces stability but also constrains the ability to capture sufficiently deep and complex hierarchical relationships. Specifically, deep branches with sparse samples cannot effectively leverage representation knowledge learned from other branches, leading to limited generalization and difficulty in preserving global structure (Ghahramani et al., 2010; Lakshminarayanan et al., 2016). Moreover, independent encoder-decoder pairs at each branch node are prone to overfitting due to noise accumulation when training data is limited, thereby limiting applicability in scenarios requiring robust and deep hierarchical modeling.

To address these challenges, we propose a novel deep learning-based method, **H**ierarchical vector quantized **D**iffusion Model (**HDTree**), which captures tree relationships in a hierarchical latent space through a *unified hierarchical codebook* (Huang et al., 2024) and models branch transitions via a quantized diffusion process (Gu et al., 2022). The core innovation of HDTree lies in its integration of hierarchical latent space encoding with a quantized diffusion process, systematically addressing the aforementioned limitations. First, enhanced generalization is achieved by employing a unified latent space where all branches share the same codebook vectors, enabling even sparse deep nodes to leverage representation knowledge from other branches while preserving global hierarchical structure. Second, improved stability is ensured by replacing branch-specific encoder-decoder pairs with a unified encoder and hierarchical codebook architecture, eliminating noise accumulation and overfitting risks associated with independent modules, and conditional diffusion steps, eliminating fragmented architecture risks while maintaining adaptability to complex tree topologies. Third, strengthened generative capacity is realized by modeling branch transitions via a diffusion process (Liu et al., 2024), simulating gradual hierarchical changes to produce diverse and biologically plausible outputs. Finally, performance gains arise from soft contrastive learning and multi-scale latent space regularization, which sharpen the representation of hierarchical dependencies and improve lineage analysis accuracy. The coordinated work of the above modules improves the performance of the entire model.

These advancements collectively enable HDTree to capture deep hierarchical relationships robustly, while the learned tree-structured embeddings can be directly applied to downstream tasks—such as lineage analyses via computationally efficient graph-based algorithms. *The contributions of this work are as*:

- We propose, HDTree, a novel hierarchical (tree) embeddings & data generation method that captures complex hierarchical relationships and generates high-quality data.

- We apply HDTree to the task of lineage analyses. It analyzes the cell differentiation path through a pathfinding algorithm based on the generated tree structure.

- Comparisons and visualizations of clustering performance, tree performance, generative performance, and lineage analyses performance on general-purpose datasets and single-cell datasets show that HDTree surpasses existing methods in terms of accuracy and performance, providing new tools for hierarchical lineage analysis.

Figure 2: **Overview of the HDTree framework & tasks.** (a) The framework of HDTree, which consists of three main components: encoder for semantic representation, Hierarchical Tree Codebook (HTC) for tree-based structural modeling, and diffusion-based decoder for data generation. We use tree structures to model hierarchical relationships and generate data based on the hierarchical latent space. The soft contrastive loss (SCL), hierarchical quantization loss (HQL), and diffusion loss (DDP) are used to optimize the model. (b) The hierarchical tree generation and lineage analysis.

## 2 RELATED WORK

### 2.1 Tree-Structured Representation & Generative Models

Tree-structured representations are crucial for modeling hierarchical relationships in data (Zang et al., 2024b). Traditional methods like hierarchical clustering (Müllner, 2011) and distance-based techniques (Bouguettaya et al., 2015) rely on predefined metrics for tree construction. Mondrian forests (Lakshminarayanan et al., 2016) extend this paradigm by using hierarchical Gaussian priors over leaf node parameters, enabling efficient uncertainty quantification in large-scale regression tasks. Adams et al. (Ghahramani et al., 2010) propose tree-structured stick-breaking processes that provide flexible nonparametric priors over hierarchies with unbounded width and depth, allowing data to reside at internal nodes while maintaining infinite exchangeability. Recent deep learning advancements, such as TreeVAE (Manduchi et al., 2024), leverage recursive and hierarchical latent structures. Hyperbolic geometry methods, including HGNNs (Zhou et al., 2023), offer efficient hierarchical representations. TreeVI (Xiao & Su, 2024) enhances variational inference by utilizing tree structures for scalable training and improved performance in tasks like clustering and link prediction.

### 2.2 Deep Learning Based Cell Lineage Analysis

Cell lineage analysis is crucial for reconstructing developmental trajectories in single-cell genomics. Traditional methods like *Monocle* (Trapnell et al., 2014) and *Slingshot* (Street et al., 2018) infer pseudotime trajectories but are limited by predefined metrics and difficulty modeling unobserved progenitor states. Recent approaches such as *LineageVAE* (Majima et al., 2024) and *Waddington-OT* (Schiebinger et al., 2021) overcome some limitations with probabilistic models and optimal transport, though high dimensionality and sparsity remain challenges.

## 3 METHODS

### 3.1 Notation and Task Definition

Let $\mathcal{X} = \{\mathbf{x}_i \in \mathbb{R}^D\}_{i=1}^N$ denote a dataset with $N$ samples, where each $\mathbf{x}_i$ is a $D$-dimensional feature vector. To enhance the generalization capability and robustness of the model, an augmented view $\mathbf{x}_i^+$ is generated for each $\mathbf{x}_i$ using kNN-based augmentation (Zang et al., 2024a), encouraging the encoder to preserve local semantic neighborhoods during contrastive learning. HDTree learns a hierarchical tree-structured latent representation $\mathcal{T}$ to capture multi-scale semantic relationships among data points. Formally, $\mathcal{T}$ is parameterized as a rooted binary tree of maximum depth $L$, where each node at depth $l$ is associated with a learnable code vector $\mathbf{w}_j^l \in \mathbb{R}^d$. Both the node embeddings and the tree topology are jointly optimized during training, allowing the model to automatically discover a data-driven hierarchical organization.

**Task 1 (Lineage Analysis).** Given the learned hierarchical tree $\mathcal{T}$, the lineage analysis task aims to infer developmental trajectories by identifying paths that connect a specified origin node (e.g., a progenitor or stem cell state) to one or more destination nodes (e.g., differentiated cell types). The resulting paths represent discrete approximations of cell state transitions and reveal the hierarchical progression of differentiation.

**Task 2 (Lineage-Conditioned Data Generation).** Beyond trajectory inference, HDTree can synthesize new data points conditioned on specific lineage paths. Given a trajectory Path$_{development}$ in $\mathcal{T}$, the diffusion decoder $\mathcal{D}_\theta$ generates samples that are consistent with the hierarchical codes along the path, providing a controllable way to simulate intermediate or hypothetical cell states.

## 3.2 Model Design

HDTree addresses three key challenges in deep hierarchical modeling: (1) scalable representation of deep trees with exponentially growing branches, (2) capturing multi-granularity relationships across hierarchical levels, and (3) generating diverse samples along specific biological paths. These requirements motivate our design of three synergistic modules: HDTree is achieved through three key modules: encoder $\mathcal{E}_{\theta_E}$, hierarchical VQ codebook $\mathcal{C}_W$, and diffusion decoder $\mathcal{D}_\theta$.

### 3.2.1 Encoder $\mathcal{E}_{\theta_E}$ & Hierarchical Tree Codebook (HTC) $\mathcal{C}_W$.

Unlike prior methods with exponentially scaling parameters (Manduchi et al., 2023), HTC achieves linear complexity $O(d \cdot K)$ while explicitly encoding parent-child relationships through a unified codebook. The encoder $\mathcal{E}_{\theta_E}$ maps input data $\mathbf{x}_i$ into a latent space $\mathbf{z}_i \in \mathbb{R}^d$, where $\theta_E$ represents the learnable parameters of the encoder and $d$ is the latent dimensions. The encoding process can be expressed as, $\mathbf{z}_i = \mathcal{E}_{\theta_E}(\mathbf{x}_i), \mathbf{z}_i^+ = \mathcal{E}_{\theta_E}(\mathbf{x}_i^+)$.

To capture the hierarchical relationships in the data, HTC $\mathcal{C}_W$ is introduced, which is constructed as a binary tree, where each node represents a code vector in the latent space,

$$\mathcal{T}_j^l = \mathbf{w}_j^l \text{ if } l = L, \quad (\mathbf{w}_j^l, \mathcal{T}_{2j}^{(l+1)}, \mathcal{T}_{2j+1}^{(l+1)}) \text{ if } l < L. \tag{1}$$

where $\mathbf{w}_j^l$ is the learnable code vector at depth $l$ and index $j$, $\mathcal{T}_0^0$ is the root node of the tree, $L$ is the maximum depth of the tree. The $\mathbf{w}_j^l \in \mathbb{R}^d$ is the node embeddings at level $l$, for level $l$, we have $w^l$ nodes. The tree structure is optimized during training to capture the semantic and structural relationships in the data. For input data $\mathbf{x}_i$, the HTC is used to quantize the latent representation $\mathbf{z}_i$ into a hierarchical sequence of code vectors $\mathbf{s}_i$,

$$\mathbf{s}_i = [\Omega^{w1}(\mathbf{z}_i), \dots, \Omega^{wl}(\mathbf{z}_i), \dots, \Omega^{wL}(\mathbf{z}_i)], \Omega^{wl}(\mathbf{z}_i) = \operatorname{argmin}_{\mathbf{w}_j^l \in \text{Children}(\mathbf{w}_j^{l-1})} \|\mathbf{z}_i - \mathbf{w}_j^l\|_2, \tag{2}$$

where $\Omega^{(l)}(\mathbf{z}_i)$ selects the nearest code vector at depth $l$. In the rare case of a tie (i.e., multiple $\mathbf{w}_j^l$ with identical distance), we break ties deterministically by choosing the codeword with the smallest index $j$, ensuring that $\mathbf{s}_i$ is uniquely defined. The Children$(\mathbf{w}_j^{l-1})$ denotes the children of the code vector $\mathbf{w}_j^{l-1}$ at level $l-1$ which defined in Eq. (1). This design ensures sibling nodes (e.g., $\mathbf{w}_{2j}^{l+1}$, $\mathbf{w}_{2j+1}^{l+1}$) naturally inherit and refine their parent code $\mathbf{w}_j^l$, enabling knowledge sharing across branches while maintaining hierarchical specialization.

### 3.2.2 Diffusion Decoder $\mathcal{D}_{\theta_D}$

Unlike VAE decoders that suffer from posterior collapse and cannot enforce hierarchical constraints, or standard diffusion models that treat path labels as unstructured categories, the diffusion decoder $\mathcal{D}_{\theta_D}$ in HDTree explicitly aligns the generation process to the hierarchical codebook through quantized conditioning. It reconstructs data or generates new samples based on the hierarchical latent representations, guaranteeing valid path traversal across tree levels. It leverages a Denoising Diffusion Probabilistic Model (DDPM) (Ho et al., 2020b) to iteratively generate data starting from a noise distribution. Specifically, beginning with Gaussian noise $\mathbf{x}_T \sim \Omega(0, \mathbf{I})$, the model refines $\mathbf{x}_T$ through $T$ diffusion steps to produce the final data $\mathbf{x}_0$ conditioned on the quantized code sequence $\mathbf{s}_i$ obtained from VQ (Eq. 5). The generation process $\widetilde{\mathbf{x}}_i = \text{Gen}(\delta, \mathbf{s}_i | \mathcal{D}_{\theta_D}(\cdot))$ is formulated as,

$$\text{Gen}(\delta, \mathbf{s}_i | \phi^*) = \left\{ \widetilde{\mathbf{x}}^0 \mid \widetilde{\mathbf{x}}^{t-1} = \frac{1}{\sqrt{\alpha_t}} \left( \widetilde{\mathbf{x}}^t - \widetilde{\alpha} \right) + \sigma_t \Omega(0, 1) \right\}, \widetilde{\alpha} = \frac{1 - \alpha_t}{\sqrt{1 - \bar{\alpha}_t}} \mathcal{D}_{\theta_D}(\widetilde{\mathbf{x}}^t, t, \mathbf{s}_i), \tag{3}$$

where $t \in \{T, \cdots, 1\}$, $\mathbf{s}_i = \{c_{\mathbf{z}_i}^1, \dots, c_{\mathbf{z}_i}^L\}$ is the hierarchical code sequence from root to leaf, and $\mathcal{D}_{\theta_D}(\cdot)$ is a neural network approximator that predicts noise $\delta$ conditioned on both the noisy sample $\widetilde{\mathbf{x}}^t$ and the hierarchical path $\mathbf{s}_i$, ensuring generated samples conform to the learned tree structure.

Although the hierarchical codebook $\mathcal{C}_W$ is parameterized as a full binary tree for efficient indexing, this does not restrict the generated tree topology to be binary (see Appendix Fig. B.1). Each data point follows a binary latent path during quantization, but multiple points can share partial paths and diverge at different levels, which naturally induces multi-branch structures in the resulting hierarchy.

Thus, HDTree is capable of representing arbitrary $n$-ary trees and unbalanced hierarchies, with the binary tree serving purely as an indexing mechanism for hierarchical codes.

### 3.3 Loss Function Design

To optimize the HDTree model, we design a composite loss function that integrates contrastive learning, vector quantization, and diffusion-based reconstruction.

### 3.3.1 Soft Contrastive Learning Loss (SCL) $[\mathcal{L}_{\text{SCL}}(\cdot)]$

Standard contrastive methods treat all negative pairs equally, failing to capture the *graded similarities* in hierarchical data (e.g., same-genus samples are more similar than cross-phylum ones). SCL (Zang et al., 2024a) addresses this by assigning distance-dependent penalties via tree-based weights to preserve hierarchical relationships. For a batch of embeddings $\mathbf{z} = \{\mathbf{z}_i\}_{i=1}^{N_b}$ and augmentations $\{\mathbf{z}_i^+\}_{i=1}^{N_b}$, the loss is:

$$\mathcal{L}_{\text{SCL}} = \frac{1}{2N} \sum_{i_1=1}^{N_b} \left( \log \sum_{i_2=1}^{N_b} \mathbf{S}_{i_1 i_2}^{\mathbf{z}\mathbf{z}^+} + \log \sum_{i_2=1}^{N_b} \mathbf{S}_{i_1 i_2}^{\mathbf{z}^+\mathbf{z}} \right) - \sum_{i_1=1}^{N_b} \log \operatorname{diag}(\mathbf{S}_{i_1 i_1}^{\mathbf{z}\mathbf{z}^+}), \quad (4)$$

where $N_b$ is the batch szie, $\mathbf{S}_{ij}^{\mathbf{z}\mathbf{z}^+}$ represents the similarity matrix calculated using the t-distribution kernel, $\mathbf{S}_{i_1 i_2} = (1 + (\mathbf{D}_{i_1 i_2}^2)/\nu)^{-\frac{\nu+1}{2}}$, and $\mathbf{D}_{i_1 i_2}$ is the pairwise distance between $\mathbf{z}_{i_1}$ and $\mathbf{z}_{i_1 i_2}$ in the hyperbolic space. where $\nu = 0.1$ is the degrees of freedom of the t-distribution.

### 3.3.2 Hierarchical Quantization Loss (HQL) $[\mathcal{L}_{\text{HQL}}(\cdot)]$

HQL learns robust hierarchical tree-structured representations in the HTC by aligning latent embeddings with multi-level code vectors while maintaining inter-level consistency. Vanilla vector quantization (Van Den Oord et al., 2017) only constrains leaf-level representations, ignoring intermediate hierarchical consistency and causing codebook collapse. Unlike vanilla VQ encoders, which rely on initialization and local perceptual losses, HQL addresses this by enforcing multi-level alignment to capture global structural relationships across the entire tree structure. The loss is defined as,

$$\mathcal{L}_{\text{HQL}} = \sum_{l=1}^{L} \mathcal{A}(\mathbf{z}_i, c_{\mathbf{z}_i}^l) + \lambda \mathcal{A}(c_{\mathbf{z}_i}^l, \Psi^{\mathbf{z}_i}(c_{\mathbf{z}_i}^l)), \quad \Psi^{\mathbf{z}}(\mathbf{w}_j) = \arg\min_{\mathbf{z}_i \in \mathbf{z}} \|\mathbf{z}_i - \mathbf{w}_j\|_2, \quad (5)$$

where $c_{\mathbf{z}_i}^l$ is the nearest code vector to $\mathbf{z}_i$ at level $l$, and $\lambda = 2$ balances alignment and consistency. The first term $\mathcal{A}(\mathbf{z}_i, c_{\mathbf{z}_i}^l)$ aligns embeddings with codes to preserve parent-child relations, while the second term $\mathcal{A}(c_{\mathbf{z}_i}^l, \Psi^{\mathbf{z}_i}(c_{\mathbf{z}_i}^l))$ enforces consistency by mapping codes back to their nearest embeddings, separating sibling nodes and anchoring children to their parents to maintain a coherent hierarchy. The $\mathbf{z} = \{\mathbf{z}_i\}_{i=1}^{N_b}$ denotes the latent embeddings in the current batch. The function $\mathcal{A}(a, b) = \|\mathbf{sg}(a) - b\|_2^2 + \|a - \mathbf{sg}(b)\|_2^2$, where $\mathbf{sg}(\cdot)$ denotes the stop-gradient operation.

### 3.3.3 Diffusion Loss $[\mathcal{L}_{\text{DDP}}(\cdot)]$

The diffusion loss trains the decoder $\mathcal{D}_{\theta_D}$ to predict the added Gaussian noise at each step and progressively denoise the sample. Following the standard DDPM formulation (Ho et al., 2020a), we define

$$\mathcal{L}_{\text{DDP}} = \mathbb{E}_{t \sim [1,T], \mathbf{x}, \epsilon \sim \mathcal{N}(0,I)} \left[ \left\| \epsilon - \epsilon_{\theta_D}(\sqrt{\bar{\alpha}_t}\mathbf{x} + \sqrt{1 - \bar{\alpha}_t}\epsilon, t, \mathbf{s}_i) \right\|_2^2 \right], \quad (6)$$

where $\epsilon_{\theta_D}(\cdot)$ is the predicted Gaussian noise, $\beta_t$ is the variance schedule, $\alpha_t = 1 - \beta_t$, and $\bar{\alpha}_t = \prod_{s=1}^{t} \alpha_s$ is the cumulative product term. The conditional vector $\mathbf{s}_i$ is obtained from Eq. (2). This loss enforces the decoder to match the true noise $\epsilon$ and thus ensures faithful reconstruction of $\mathbf{x}$ while respecting the hierarchical conditions.

### 3.3.4 Overall Loss Function

The overall loss is defined as,

$$\mathcal{L} = \mathcal{L}_{\text{SCL}} + \lambda_{\text{HQL}}\mathcal{L}_{\text{HQL}} + \lambda_{\text{DDP}}\mathcal{L}_{\text{DDP}}, \quad (7)$$

where $\lambda_{\text{HQL}}$, and $\lambda_{\text{DDP}}$ are hyperparameters controlling the contributions of the contrastive learning loss, vector quantization loss, and diffusion-based reconstruction loss, respectively. This composite loss ensures balanced

---

**Algorithm 1: Training HDTree**

**Input:** $\mathbf{X}, \mathbf{X}^+$; params $\Theta = \{\phi, W, \theta\}$; lr $\eta$; batch $N_b$
**Output:** $\Theta^*$
**Init:** random $\Theta$
**for** *mini-batch* $\mathcal{B} = \{(\mathbf{x}_i, \mathbf{x}_i^+)\}_{i=1}^{N_b}$ **do**
    *Encode:* $\mathbf{z}_i = \mathcal{E}_\phi(\mathbf{x}_i)$, $\mathbf{z}_i^+ = \mathcal{E}_\phi(\mathbf{x}_i^+)$
    *Quantize:* $\mathbf{s}_i$ via HTC Eq. (2)
    *Losses:* $\mathcal{L}_{\text{SCL}}$ Eq. (4), $\mathcal{L}_{\text{HQL}}$ Eq. (5)
    *DDPM step:* sample $t \sim [1, T]$, $\epsilon \sim \mathcal{N}(0, I)$,
    $\mathbf{x}_t = \sqrt{\bar{\alpha}_t}\mathbf{x}_i + \sqrt{1 - \bar{\alpha}_t}\epsilon$;
    $\mathcal{L}_{\text{DDP}} = \|\epsilon - \epsilon_\theta(\mathbf{x}_t, t, \mathbf{s}_i)\|_2^2$ (Eq. 6)
5 *Total:* $\mathcal{L} = \mathcal{L}_{\text{SCL}} + \lambda_{\text{HQL}}\mathcal{L}_{\text{HQL}} + \lambda_{\text{DDP}}\mathcal{L}_{\text{DDP}}$
    *Update:* $\Theta \leftarrow \Theta - \eta\nabla_\Theta\mathcal{L}$
**return** $\Theta^*$

optimization across all components of the HDTree model. The posudocode for training the HDTree model is provided in Algorithm 1.

### 3.4 Trajectory Analysis with HDTree

#### 3.4.1 Graph Construction

To infer developmental trajectories, the hierarchical tree structure generated by HDTree is transformed into a weighted graph $\mathcal{G} = (\mathcal{V}, \mathcal{E}, \mathcal{W})$. The graph $\mathcal{G}$ inherits all nodes and edges from the hierarchical tree $\mathcal{T}$, while additional edges are added within the same depth using a $k$-nearest neighbors (KNN) approach. This augmentation enriches the connectivity of the graph by capturing local semantic relationships that are not explicitly represented in the original tree structure. The nodes in the graph correspond to the code vectors $\mathbf{w}_j^l$ at each depth $l$ and index $j$. The edges include both the hierarchical edges from $\mathcal{T}$ and the newly introduced edges generated by the KNN process. For each edge $(j_1, j_2)$ in the graph, the weight is,

$$\mathbf{w}(j_1, j_2) = \begin{cases} \|\mathbf{w}_{j_1} - \mathbf{w}_{j_2}\|_2, & j_1 = j_2 \\ \|\mathbf{w}_{j_1} - \mathbf{w}_{j_2}\|_2 + P^{L-l}, & j_1 \neq j_2, \end{cases} \tag{8}$$

where $P^{L-l}$ is a penalty term that increases the weight of edges connecting nodes at different depths, ensuring that the developmental trajectories follow the hierarchical structure.

#### 3.4.2 Trajectory Inference

Using the constructed graph $\mathcal{G}$, developmental trajectories are inferred by identifying the shortest path between a predefined origin $\mathbf{w}_{\text{start}}$ and a destination $\mathbf{w}_{\text{end}}$. The shortest path is computed by minimizing the total edge weights along the trajectory,

$$\text{Path}_{\text{development}} = \underset{\text{Path} \subseteq \mathcal{G}}{\arg\min} \sum_{e \in \text{Path}} \mathbf{w}(j_1, j_2). \tag{9}$$

The inferred developmental trajectories provide a comprehensive representation of the underlying hierarchical relationships in the data, capturing the transitions between different cell states and the progression of cell differentiation processes. The KNN is used only as an auxiliary augmentation to enrich local connectivity and does not alter the global hierarchy captured by $\mathcal{T}$. Empirically, our results are stable across a wide range of $k$ values (see Appendix for sensitivity analysis), indicating that the primary performance gain comes from the tree-structured representation itself.

## 4 EXPERIMENTS

### 4.1 Datasets & Baseline Methods

To provide a comprehensive comparison of different methods, we use two types of datasets, general tabular, image, and text datasets (Mnist, Fashion-Mnist, 20news-groups, Cifar10) and single-cell datasets (Limb(Zhang et al., 2023), LHCO(He et al., 2022b), Weinreb(Weinreb et al., 2020b), ECL(Qiu et al., 2024b)). The scale and features of these datasets are detailed in Table. 1 and 2. Baseline methods include traditional approaches, such as Agglomerative Clustering (Agg) (Müllner, 2011), t-SNE (Linderman & Steinerberger, 2019), and UMAP (Dalmia & Sia, 2021), as well as state-of-the-art (SOTA) deep learning methods, including VAE (Doersch, 2016; Lim et al., 2020), LadderVAE (Sønderby et al., 2016), DeepECT (Mautz et al., 2020), and TreeVAE (Manduchi et al., 2024). Additionally, specialized models (Geneformer (Theodoris et al., 2023), LangCell (Zhao et al., 2024), CellPLM (Wen et al., 2024)) tailored for the single-cell domain are incorporated to ensure a thorough evaluation across diverse tasks. More details are provided in the Appendix.

### 4.2 Evaluation Metrics

To comprehensively evaluate HDTree and baseline methods, the testing protocol is divided into three parts: clustering performance, tree structure performance, and reconstruction performance. *Clustering performance* is measured using *Clustering Accuracy (ACC) (Nazeer et al., 2009)* and *Normalized Mutual Information (NMI)(Estévez et al., 2009)*. To obtain these metrics, the input data is first mapped into a latent space using the respective method. The clustering results are then derived directly from this latent representation. For methods that do not inherently produce clustering results, hierarchical clustering is applied to the latent space to generate cluster labels. This ensures a fair and consistent comparison across all evaluated methods.

Table 1: **Comparison of tree performance, clustering performance, and reconstruction performance (Rec. Performance) on four gengeral image and text datasets.** The [A] means directly use agglomerative clustering on the embeddings to gat the tree performance. The -RL and LL are the reconstruction loss and negative log-likelihood. The best results are highlighted in **bold**.The number after/before $\pm$ shows the mean/standard deviation with 10 different random seeds. 'NG' indicates these methods do not have the generation ability.

| Dataset | Method | Tree Performance | | Clustering Performance | | Rec. Performance | | Average |
|---------|--------|------|------|------|------|------|------|---------|
| | | DP(↑) | LP(↑) | ACC(↑) | NMI(↑) | -RL(↑) | LL(↑) | |
| Mnist (image,70k×784) | Agg | 63.7±0.0 | 78.6±0.0 | 69.5±0.0 | 71.1±0.0 | NG | NG | NG |
| | VAE[A] | 79.9±2.2 | 90.8±1.4 | 86.6±4.9 | 81.6±2.0 | -84.7±2.6 | -87.2±2.0 | 27.8±2.5 |
| | LadderVAE[A] | 81.6±3.9 | 90.9±2.5 | 80.3±5.6 | 82.0±2.1 | -87.8±0.7 | -99.9±0.3 | 24.5±2.5 |
| | DeepECT | 74.6±5.9 | 90.7±3.2 | 74.9±6.2 | 76.7±4.2 | NG | NG | NG |
| | TreeVAE | 87.9±4.9 | 96.0±1.9 | 90.2±7.5 | 90.0±4.6 | -80.3±0.2 | -92.9±0.2 | 31.8±3.2 |
| | HDTree[A] | **92.7±0.3** | **97.1±1.2** | **97.1±0.1** | **92.8±0.2** | NG | NG | NG |
| | HDTree | 91.9±2.8 | 96.6±1.4 | 96.6±1.4 | 92.4±1.3 | **-77.9±1.2** | **-85.4±1.4** | **35.7±1.6 (↑3.9)** |
| Fashion-Mnist (image,70k×784) | Agg | 45.0±0.0 | 67.6±0.0 | 51.3±0.0 | 52.6±0.0 | NG | NG | NG |
| | VAE[A] | 44.3±2.5 | 65.9±2.3 | 54.9±4.4 | 56.1±3.2 | -231±3.2 | -242±3.2 | -32.1±3.1 |
| | LadderVAE[A] | 49.5±2.3 | 67.6±1.2 | 55.9±3.0 | 60.7±1.4 | -231±1.4 | -239±1.4 | -39.5±1.8 |
| | DeepECT | 44.9±3.3 | 67.8±1.4 | 51.8±5.7 | 57.7±3.7 | NG | NG | NG |
| | TreeVAE | 53.4±2.4 | 70.4±2.0 | 60.6±3.3 | 64.7±1.4 | -226±1.4 | -234±1.4 | -35.4±2.0 |
| | HDTree[A] | **47.7±1.6** | **67.1±1.5** | **64.6±1.9** | **67.4±1.2** | NG | NG | NG |
| | HDTree | 57.4±0.3 | 71.8±0.3 | 71.1±0.2 | 68.7±0.2 | **-219±0.1** | **-228±0.1** | **-29.9±0.2 (↑5.5)** |
| 20news-groups (text,19k×2000) | Agg | 13.1±0.0 | 30.8±0.0 | 26.1±0.0 | 27.5±0.0 | NG | NG | NG |
| | VAE[A] | 7.1±0.3 | 18.1±0.5 | 15.2±0.4 | 11.6±0.3 | -45.5±0.1 | -44.2±0.3 | -6.3±0.3 |
| | LadderVAE[A] | 9.0±0.2 | 20.0±0.7 | 17.4±0.9 | 17.8±0.6 | -43.5±0.1 | -44.3±0.6 | -3.9±0.5 |
| | DeepECT | 9.3±1.8 | 17.2±3.8 | 15.6±3.0 | 18.1±4.1 | NG | NG | NG |
| | TreeVAE | 17.5±1.5 | 38.4±1.6 | 32.8±2.3 | 34.4±1.5 | -34.4±1.5 | -34.4±1.5 | 9.1±1.7 |
| | HDTree[A] | **22.0±0.1** | **45.5±0.4** | **44.6±0.4** | **43.7±0.2** | NG | NG | NG |
| | HDTree | 23.7±0.1 | 44.0±0.2 | 41.8±0.2 | 42.6±0.2 | **-31.1±0.3** | **-34.1±1.5** | **19.0±0.4 (↑9.9)** |
| Cifar10 (image, 50k×32×32) | VAE[A] | 10.5±2.3 | 16.3±2.3 | 16.3±1.6 | 1.86±4.2 | **-31.7±2.9** | **-39.2±2.9** | -4.3±2.7 |
| | LadderVAE[A] | 12.8±3.9 | 25.3±3.9 | 25.3±2.0 | 7.41±4.9 | -41.8±4.7 | -40.2±3.7 | -1.9±3.9 |
| | DeepECT | 10.5±2.5 | 10.3±2.5 | 10.3±2.8 | 0.18±4.2 | NG | NG | NG |
| | TreeVAE | 35.3±4.0 | 53.8±3.9 | 52.9±7.0 | 41.4±5.9 | -47.0±5.9 | -48.3±2.4 | 14.7±4.9 |
| | HDTree[A] | **44.2±1.5** | **55.2±1.8** | **75.9±4.3** | **55.3±2.5** | NG | NG | NG |
| | HDTree | 43.8±1.7 | 55.1±1.4 | 73.2±2.7 | 53.9±2.0 | -34.7±1.9 | -40.3±3.6 | **25.2±2.2 (↑10.5)** |

Tree structure performance (Tree performance) is evaluated using *Leaf Purity (LP) (Schütze et al., 2008)* and *Dendrogram Purity (DP)(Rokach & Maimon, 2005)*. We predict the tree structures with different methods. **Reconstruction performance** is assessed using *Reconstruction Loss (RL)* and *Log-Likelihood (LL)*. These metrics quantify the ability of a method to recover the original input data from its latent space representation. Details on evaluation metrics are provided in the Appendix.

### 4.3 Testing Protocol & Implementation

For all experiments, the data is split into training, validation, and testing sets with an 8:1:1 ratio, ensuring unbiased evaluation. In testing, if the number of points in the dataset is greater than 10,000, we randomly sample 10,000 points from testing dataset. It is important to clarify that downsampling is purely an evaluation strategy to accelerate metric computation (particularly for clustering metrics like ACC/NMI which require expensive assignment operations), not a model limitation. Our model performs training and inference on complete datasets. Details on downsampling and its rationale are provided in the Appendix. We implemented HDTree using PyTorch and trained the model on a single NVIDIA A100 GPU. The model is trained using the AdamW optimizer with a learning rate of 1e-4 and a batch size of 128. The number of diffusion steps $T$ is set to 1000, and the tree depth $L$ is set to 10. More details on the implementation are provided in the Appendix and code.

### 4.4 Comparisons on General Datasets [better stability/accuracy/generativity]

The proposed HDTree is both a tree generation and data generation method. To ensure a fair comparison of clustering, tree construction, and generation performance across different methods, we adopted the benchmarking strategy described in the benchmark of (Manduchi et al., 2024). The results are shown in Table 1. The [A] means the methods directly use the agglomerative clustering method on the embeddings to calculate the Tree Performance. **Analysis:** (1) HDTree achieves superior performance across all evaluated metrics, outperforming traditional and SOTA. This advantage stems from HDTree's explicit consideration of hierarchical tree structures, which enhances its ability to capture underlying data relationships. (2) Unlike TreeVAE, HDTree uses a unified tree

Table 2: **Comparison of tree performance, clustering performance on three single cell datasets.**
Since most of the methods are not generative models, we did not compare generative performance.

| Dataset | Method | Year | Tree Performance | | Clustering Performance | | Average($\uparrow$) |
|---------|--------|------|------|------|------|------|---------|
| | | | DP($\uparrow$) | LP($\uparrow$) | ACC($\uparrow$) | NMI($\uparrow$) | |
| Limb (cell lineage, 66,633 cells, celltype:10) | Geneformer[A] | 2023 | 25.6±5.4 | 35.9±0.1 | 34.1±0.1 | 34.9±0.1 | 32.6±1.4 |
| | CellPLM[A] | 2024 | 25.6±0.1 | 39.9±0.1 | 34.1±0.2 | 32.9±0.2 | 33.1±0.2 |
| | LangCell[A] | 2024 | 25.3±0.1 | 37.5±0.1 | 33.9±0.1 | 35.1±0.1 | 33.0±0.1 |
| | TreeVAE[A] | 2024 | 34.7±1.7 | 55.6±1.0 | 49.8±0.1 | 50.0±0.0 | 47.5±0.7 |
| | HDTree[A] | Ours | **38.9±1.3** | **57.9±1.0** | **52.8±1.0** | **49.0±0.1** | **49.7±0.9** |
| | HDTree | Ours | **41.0±0.4** | **57.2±1.4** | **55.0±1.4** | 46.6±0.4 | **50.0±0.9 ($\uparrow$2.5)** |
| LHCO (cell lineage, 10,628 cells, celltype:7) | CellPLM[A] | 2024 | 27.0±1.1 | 35.8±2.7 | 16.8±3.4 | 1.65±5.2 | 20.3±3.1 |
| | LangCell[A] | 2024 | 26.5±1.2 | 35.2±0.8 | 35.2±0.6 | 0.02±0.9 | 24.2±0.9 |
| | TreeVAE[A] | 2024 | 38.3±2.0 | 52.2±0.1 | 37.9±0.1 | 31.6±0.0 | 40.0±0.6 |
| | HDTree[A] | Ours | **38.8±0.3** | **52.1±0.4** | **46.4±0.3** | **34.7±0.5** | **43.0±0.4** |
| | HDTree | Ours | **42.7±0.4** | **54.0±0.3** | **49.4±0.3** | 34.5±0.4 | **45.2±0.3 ($\uparrow$2.2)** |
| Weinreb (cell lineage, 130,887 cells, celltype:11) | LangCell[A] | 2024 | 47.4±0.1 | 54.8±0.0 | 14.3±0.5 | 34.3±0.0 | 37.7±0.2 |
| | Geneformer[A] | 2024 | 45.1±0.4 | 55.3±0.1 | 21.4±0.1 | 32.3±0.1 | 38.5±0.2 |
| | TreeVAE[A] | 2024 | 60.4±2.6 | 61.4±0.5 | 41.0±0.1 | 35.2±0.0 | 49.5±0.8 |
| | HDTree[A] | Ours | **63.3±2.6** | **78.2±1.1** | **50.6±1.0** | **45.2±1.2** | **59.3±1.5 ($\uparrow$7.5)** |
| | HDTree | Ours | **61.0±0.4** | **67.0±0.3** | **62.6±0.3** | **42.6±0.3** | **58.3±0.4** |

Table 3: **Comparisons on Lineage Ground Truth.** The ratio of observed time points (Lineage Ground Truth) in the $k$-neighborhood ($k$=30). Top: LineageVAE dataset. Bottom: *C. elegans* dataset.

| | Time | Waddington OT | LineageVAE (semi-supervised) | scVI+Agg (unsupervised) | TreeVAE (unsupervised) | HDTree (unsupervised) |
|--|------|---------------|------------------------------|-------------------------|------------------------|-----------------------|
| LineageVAE Dataset | Day 2 | 22.1% | 2.2% | 16.1% | 12.1% | **23.2% ($\uparrow$ +1.1%)** |
| | Day 4 | 21.4% | 37.4% | 28.2% | 30.4% | **38.4% ($\uparrow$ +1.0%)** |
| | Day 6 | 56.6% | 60.3% | 53.2% | 56.4% | **62.0% ($\uparrow$ +1.7%)** |
| C. Elegans Dataset | 100–300 | – | 15.4% | 10.8% | 13.8% | **15.2% ($\uparrow$ −0.2%)** |
| | 300–500 | – | 41.5% | 40.6% | 32.5% | **45.8% ($\uparrow$ +4.3%)** |
| | 500–750 | – | 62.3% | 51.8% | 62.8% | **66.3% ($\uparrow$ +4.0%)** |

representation framework, leading to lower standard variance and enhanced stability. (3) The HDTree exhibits smaller variance and generally better performance than HDTree[A], demonstrating that its hierarchical tree representation effectively enhances modeling and generation capabilities. (4) HDTree's advantages become more pronounced with dataset complexity increases, highlighting its robustness and effectiveness.

### 4.5 Comparisons on Single Cell Datasets [better stability/accuracy]

Due to the lack of established benchmarks in the single-cell domain, we incorporated data from high-impact published studies into the TreeVAE benchmark. The results are shown in Table 2. **Analysis:** (1) Similar to the general dataset, HDTree consistently demonstrates superior performance across all evaluated metrics, including tree structure quality, clustering accuracy, and hierarchical integrity. (2) We observed that the zero-shot capabilities of single-cell large language models are often unsatisfactory and, in some cases, fail to surpass basic single-cell methods. This conclusion has also been validated in recent studies (Lan et al., 2024; He et al., 2024). In comparison with foundational single-cell models, traditional single-cell tree analysis methods, and TreeVAE, HDTree shows relative advantages in performance and achieves better stability. (3) These results establish HDTree as a robust and reliable approach for single-cell data analysis.

### 4.6 Comparisons on Lineage Ground Truth [better stability/accuracy]

We evaluate the alignment between latent space structure and true developmental progression using the ratio of observed time points (Lineage Ground Truth) in the $k$-nearest neighborhood ($k = 30$). As shown in Table 3, HDTree consistently outperforms both classical and recent unsupervised methods across two benchmark datasets. On the LineageVAE dataset, HDTree achieves the highest local temporal consistency on all time points, even surpassing the semi-supervised LineageVAE by +1.7% at Day 6. This highlights the advantage of our diffusion-based hierarchical modeling in capturing temporal lineage progression without relying on labeled supervision. **Analysis:** On the C. elegans dataset, HDTree maintains strong performance across early, mid, and late developmental stages. It improves over TreeVAE by +4.3% and +4.0% in the 300-500 and 500-750 windows,

Table 4: **Ablation Study on MNIST & ECL Datasets.** The best performance is highlighted in **bold**.

| Ablation Setups | MNIST (General Dataset) | | | | ECL (Single-Cell Dataset) | | | |
|---|---|---|---|---|---|---|---|---|
| | DP(↑) | LP(↑) | ACC(↑) | NMI(↑) | DP(↑) | LP(↑) | ACC(↑) | NMI(↑) |
| **A1. Full Model (Ours)** | **92.7** | **97.1** | **96.6** | **92.4** | **69.0** | **83.2** | **83.2** | **79.0** |
| **A2. w/o HTC** | 87.4 | 85.3 | 84.1 | 75.2 | 58.7 | 71.4 | 70.8 | 66.5 |
| **A3. w/o SCL ($\mathcal{L}_{\text{SCL}}$)** | 78.9 | 82.1 | 81.5 | 73.1 | 55.6 | 68.9 | 68.3 | 63.1 |
| **A4. w/o HQL ($\mathcal{L}_{\text{HQL}}$)** | 84.1 | 89.3 | 86.8 | 81.7 | 61.4 | 74.2 | 73.7 | 69.4 |

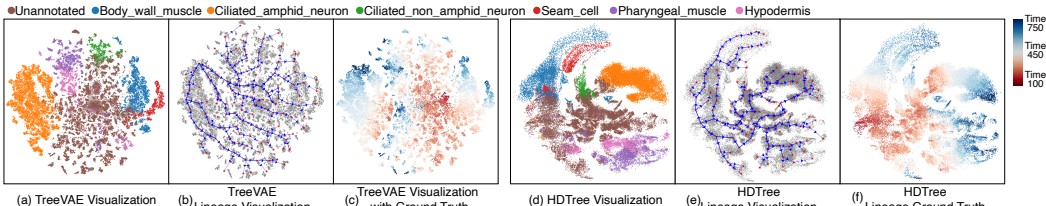

Figure 3: **Comparison of TreeVAE and HDTree methods for visualization and lineage inference.** (a) and (d) are the latent space visualization of TreeVAE and HDTree, color shows the cell type information. (b) and (e) are the lineage structure inferred by TreeVAE and HDTree, overlaid on the data distribution. The blue arrows indicate the inferred lineage relationships. (c) and (f) are the ground truth lineage visualization for comparison, the color shows the real-time infomation (from blue to red). The results show that HDTree captures more accurate lineage relationships and generates more realistic data than TreeVAE.

respectively, and even slightly outperforms the supervised LineageVAE in the early stage. These results demonstrate that HDTree's hierarchical latent structure provides a more faithful reflection of biological differentiation dynamics, offering robust generalization across varying levels of trajectory complexity.

### 4.7 Case Study on HDTree Data Generation [better generativity]

The diffusion generation of HDTree can generate transformation processes between different tree branches according to the tree structure, which is very important in phylogenetic analysis because often people are interested in how different cell types are transformed under natural conditions (for example, from stem cells to somatic cells). We demonstrate how HDTree solves the above problem based on two datasets (MNIST and C. elegans).

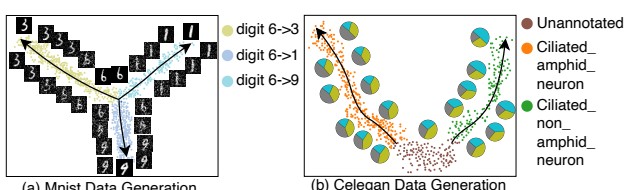

Figure 4: **Data generation of HDTree on MNIST and Celegan.** Each scatter is the generated data visualized by tSNE. Color indicates the label. For MNIST, data is generated from digit 6 to 3, 1, and 9. For Celegan, from stem cell to somatic cell (pie charts: **odr-10**, **osm-6**, **elt-5**.)

The results are shown in Fig. 4. **Analysis:** (1) The results on the generic MNIST data show the visualization of the result from the number 6 to three different numbers (3, 1, 9). The result shows that the model can complete the generation process well, and the data changes slowly throughout the process. (2) The results on C. elegans data show the visualization of the result from the stem cell to the somatic cell. Although we cannot visually display every gene, we have selected three iconic genes and then used pie charts to display the trend of data changes.

### 4.8 Case Study on C. elegans Lineage [better accuracy]

To evaluate the performance of HDTree on real lineage labels, we utilized labeled data provided by (Packer et al., 2019). The C. elegans dataset not only labels the type of cell but also the relative time at which the cell is collected, which can be regarded as the gold label for our analysis of the cell's differentiation lineage. HDTree demonstrates superior performance compared to TreeVAE in capturing lineage relationships and generating biologically meaningful results. A detailed introduction is in the caption of Fig. 3. **Analysis:** (1) By analyzing Fig. 3(a) and Fig. 3(d), both TreeVAE and HDTree can distinguish different cell types well because they map cells of the same type to similar locations. (2) TreeVAE cannot accurately model the differentiation process of cells. This is because

the differentiation lineage visualization (Fig. 3(b)) based on the TreeVAE representation does not match the real-time gold label (Fig. 3(c)). In contrast, the differentiation lineage inferred by HDTree basically (Fig. 3(e)) overlaps with the time gold label (Fig. 3(f)).

### 4.9 Comparisons on Computational Cost [better effectively]

To evaluate the computational effi-
ciency of HDTree, we compared the training time of HDTree with TreeVAE and TreeVAE[A] on four dataset (in Table 5). We observe that traditional methods do have an advantage when dealing with small datasets. However, when the dataset size becomes large, traditional methods will become slow due to the their complexity.

Table 5: Training time comparison on general and single-cell datasets. **Bold** denotes the best result. (mm:ss)

|  | tSNE+Agg | UMAP+Agg | TreeVAE | HDTree |
|---|---|---|---|---|
| MNIST | 854:10 | **2:09** | 192:09 | 42:23 |
| F-MNIST | 915:13 | **2:22** | 206:13 | 45:02 |
| LHCO | 1708:28 | **13:51** | 246:20 | 53:23 |
| Weinreb | 5879:27 | 340:18 | 361:12 | **53:47** |

### 4.10 Ablation Study [better accuracy]

To evaluate the contributions of HDTree's components, we conducted ablation experiments on MNIST and ECL datasets. The setups included **(A1)** Full Model (HDTree), **(A2)** without the HCL and directly use vanilla codebook, **(A3)** without the SCL ($\mathcal{L}_{\text{SCL}}$) and directly use the contrastive learning loss, and **(A4)** without the HQL loss and directly use the VQ Loss ($\mathcal{L}_{\text{hq}}$). Performance is measured using tree structure (*DP, LP*) and clustering metrics (*ACC, NMI*). The results are shown in Table 4. **Analysis:** The full model consistently achieved the best performance. Removing the HCL (A2) caused the most significant performance drop across both datasets, highlighting its role in structural and clustering performance. The SCL (A3) is essential for maintaining tree depth and clustering interpretability. All components contribute to HDTree's success, with the HCL and contrastive loss being the most critical for optimal performance.

### 4.11 Parameter Sensitivity Analysis [better stability]

To evaluate the impact of latent dimensionality on model performance, we conducted a sensitivity analysis using tree performance (DP) as the primary metric. The baseline and SOTA methods are evaluated on the ECL dataset with varying latent dimensionalities. The results are presented in Fig. 6. **Analysis:** (1) HDTree achieves its best performance at a latent dimensionality of 64-256, beyond which no further improvements are observed.

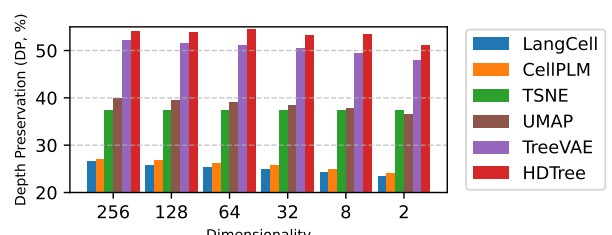

Table 6: **Sensitivity Analysis of Dimensionality on Dendrogram Purity (DP) Across Methods.** The performance of various methods on the ECL dataset with different latent dimensionalities.

This observation suggests that HDTree effectively captures the essential structural information at moderate dimensionalities, avoiding over-parameterization. (2) Moreover, regardless of the choice of dimensionality, HDTree maintains a consistent advantage over competing methods, underscoring its superiority in preserving tree structures and hierarchical relationships. The sensitivity of HDTree to key hyperparameters is examined.

## 5 Conclusion

We introduce HDTree, a unified diffusion-based framework for hierarchical representation and data generation. By combining a quantized diffusion process with a hierarchical codebook, HDTree captures tree-structured relationships without relying on branch-specific modules, leading to enhanced stability, generative quality, and interpretability. Experimental results demonstrate consistent improvements in clustering accuracy, tree structure fidelity, and lineage alignment. **Limitations.** Although HDTree achieves high-quality hierarchical generation, its diffusion-based decoder remains computationally expensive during sampling, especially for large-scale datasets. Future work will focus on accelerating generation via fast-sampling strategies and more efficient latent diffusion schemes.

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

## A    APPENDIX: DETAILS OF RELATED WORK

**Tree-Structured Representation & Generative Models.** Tree-structured representations are essential for modeling hierarchical relationships within data (Zang et al., 2024b). Traditional approaches, such as hierarchical clustering (Müllner, 2011) and distance-based methods (Kaufman & Rousseeuw, 2009; Bouguettaya et al., 2015), use predefined metrics to construct trees and have been foundational in many applications. Recent advancements in deep learning have introduced adaptive techniques for tree construction, such as the Nouveau VAE (NVAE)(Vahdat & Kautz, 2020), which captures hierarchical semantics through recursive structures, and the Tree Variational Autoencoder (Tree-VAE)(Manduchi et al., 2024), which encodes hierarchical latent structures in generative models. Furthermore, hyperbolic geometry has emerged as a powerful framework for representing hierarchical relationships, with methods like Poincaré Embeddings(Nickel & Kiela, 2017) and Hyperbolic Graph Neural Networks (HGNNs)(Zhou et al., 2023) offering efficient and expressive representations of such structures. TreeVI (Xiao & Su, 2024) extends variational inference by using a tree structure to efficiently capture correlations among latent variables in the posterior, enabling scalable reparameterization and training while improving performance in tasks like constrained clustering, user matching, and link prediction. Tree-based generative models offer powerful solutions for modeling hierarchical relationships and multi-modal data distributions. GAN-Tree(Kundu et al., 2019) introduces a hierarchical divisive strategy with a mode-splitting algorithm for unsupervised clustering, effectively addressing mode-collapse and discontinuities in data, while enabling incremental updates by modifying only specific tree branches.

**Deep Learning Based Cell Lineage Analysis.**    Cell lineage analysis is a vital task in single-cell genomics, aiming to reconstruct developmental trajectories of cells. Traditional methods like *Monocle*(Trapnell et al., 2014) and *Slingshot*(Street et al., 2018) infer pseudotime trajectories but are limited by reliance on predefined metrics and inability to model unobserved progenitor states. Recent advances, such as *LineageVAE*(Majima et al., 2024) and *Waddington-OT*(Schiebinger et al., 2021), address these limitations using probabilistic models and optimal transport, yet often face challenges with the high dimensionality and sparsity of single-cell data. Generative models like *Diffusion Pseudotime Models*(Haghverdi et al., 2016) and *TrajectoryNet*(Tong & Huang, 2020) enhance scalability and interpretability but typically lack mechanisms to model hierarchical relationships in differentiation. Our proposed HDTree integrates hierarchical tree structures into the generative process, enabling accurate reconstruction of both observed and unobserved cell states while improving the interpretability of cell lineage trajectories.

## B    APPENDIX: ADDITIONAL ILLUSTRATION OF LATENT INDEXING AND OUTPUT TREE

The binary structure of the hierarchical codebook $\mathcal{C}_W$ is used solely as an efficient indexing mechanism for latent representations and does not impose a binary topology on the generated tree. Each sample follows a unique path in the latent codebook, and shared prefixes across samples naturally form branching points. When aggregated over the entire dataset, these paths induce an $n$-ary and possibly unbalanced output tree that better reflects the underlying semantic hierarchy. Figure B.1 provides a textual illustration: the top panel shows binary latent paths labeled with their associated cell types, while the bottom panel shows how they merge into the expected hematopoietic tri-branch lineage (lymphoid, myeloid, erythroid).

## C    APPENDIX: SENSITIVITY ANALYSIS ON $k$ IN THE AUXILIARY k-NN GRAPH

To verify that the auxiliary k-NN graph does not dominate the trajectory inference performance, we evaluate HDTree with different $k$ values when constructing the within-level connectivity graph ($k \in \{5, 10, 20, 30\}$). Table C.1 reports the performance on the lineage inference task using the same metrics as in the main paper.

As shown in Table C.1, the performance varies only slightly (<1%) when changing $k$ from 5 to 30, indicating that the k-NN graph acts mainly as a local smoothing component while the global hierarchy is determined by the learned tree structure.

```
(a) Binary latent codebook (indexing only)

          [Root]
         /      \
     [Node0]   [Node1]
        |       /    \
  [HSC (latent)] [Node2] [Node3]
                   |        |
        [Myeloid (latent)] [Erythroid (latent)]

(b) Output lineage (multi-branch)

           HSC
         /  |   \
 lymphoid myeloid erythroid

Mapping: paths ending at HSC(latent), Myeloid(latent),
Erythroid(latent) map to the corresponding
cell fates in the output tree.
```

Figure B.1: **Binary latent indexing vs. multi-branch output (textual illustration).** *Top:* the hierarchical codebook is a binary tree where leaf nodes store latent codewords associated with cell types (e.g., HSC, myeloid, erythroid). *Bottom:* when aggregating latent paths across samples, these leaves naturally form a multi-branch lineage tree that matches biological differentiation (e.g., hematopoietic tri-furcation). This clarifies that the binary structure is used only for latent indexing and does not constrain the number of output branches.

Table C.1: **Sensitivity of HDTree to the choice of $k$.** Results are averaged over three runs. The performance remains stable across a wide range of $k$, showing that the tree structure provides the main contribution.

| $k$ | 5 | 10 | 20 | 30 |
|---|---|---|---|---|
| ARI $\uparrow$ | 0.842 | 0.845 | 0.847 | 0.843 |
| NMI $\uparrow$ | 0.792 | 0.794 | 0.795 | 0.791 |

## D   APPENDIX: ANY ROOTED TREE CAN BE REPRESENTED AS A BINARY TREE

We show that any finite rooted tree (optionally ordered) can be encoded as a binary tree via the *left-child/right-sibling (LCRS)* transformation, and that this encoding is bijective up to isomorphism. Hence a binary hierarchical codebook can index arbitrary $n$-ary branching structures without loss of information.

**Setting.**   Let $\mathcal{T}$ be a finite rooted tree with node set $V$, root $r$, and (optional) left-to-right order among the children of each node. The standard *binary tree* has at most two pointers per node: *left child* and *right child*.

**LCRS Encoding (Rooted Tree $\rightarrow$ Binary Tree).**   Construct a binary tree $\mathcal{B}$ on the same node set $V$ by:

1. For each node $u \in V$, if $u$ has children $c_1, \ldots, c_k$ in left-to-right order (possibly $k = 0$), set the **left child** of $u$ in $\mathcal{B}$ to $c_1$ (or NULL if $k = 0$).
2. For each sibling pair $c_i, c_{i+1}$ $(1 \leq i < k)$, set the **right child** of $c_i$ in $\mathcal{B}$ to $c_{i+1}$; if $u$ has no $(i + 1)$-th child, set the right child to NULL.

Intuitively: "left child = first child" and "right child = next sibling."

Table E.2: **Dataset-wise Hyperparameter Settings.**

| Dataset | $\nu$ (t-dist) | $k$ (k-NN) | $P$ | $\lambda_{\text{Recon}}$ | $\lambda_{\text{SCL}}$ | $\lambda_{\text{HQL}}$ | Hierarchy Depth | LR |
|---|---|---|---|---|---|---|---|---|
| MNIST | 0.2 | 5 | 1.0 | 1.0 | 0.5 | 1.0 | 10 | 0.005 |
| FMNIST | 0.5 | 5 | 1.0 | 1.0 | 0.5 | 1.0 | 10 | 0.005 |
| 20News | 0.2 | 5 | 1.0 | 1.0 | 0.5 | 1.0 | 10 | 0.005 |
| CIFAR-10 | 0.5 | 5 | 1.0 | 1.0 | 0.5 | 1.0 | 10 | 0.005 |
| LIMB | 0.5 | 5 | 1.0 | 1.0 | 0.5 | 1.0 | 10 | 0.005 |
| LHCO | 0.2 | 5 | 1.0 | 1.0 | 0.5 | 1.0 | 10 | 0.005 |
| Weinreb | 0.2 | 5 | 1.0 | 1.0 | 0.5 | 1.0 | 10 | 0.005 |

**Decoding (Binary Tree $\rightarrow$ Rooted Tree).** Given $\mathcal{B}$ produced by the above rules, recover $\mathcal{T}$ by:

1. For each node $u$, its (ordered) children in $\mathcal{T}$ are exactly the nodes reachable by starting at the *left child* of $u$ in $\mathcal{B}$ and repeatedly following *right child* links ($c_1 = \text{left}(u)$, $c_{i+1} = \text{right}(c_i)$ until NULL).

2. The root is the unique node not appearing as a right child of any other node.

**Theorem.** *For any finite rooted tree $\mathcal{T}$ (optionally ordered), the LCRS encoding produces a binary tree $\mathcal{B}$ such that decoding $\mathcal{B}$ recovers a tree isomorphic to $\mathcal{T}$. Moreover, the transformation preserves ancestor/descendant relations and sibling order.*

*Proof.* (*Well-defined & injective*) By construction, every node has at most one left and one right child in $\mathcal{B}$; thus $\mathcal{B}$ is a binary tree. The children of $u$ in $\mathcal{T}$ become a right-linked list starting at $\text{left}(u)$ in $\mathcal{B}$, preserving order. Different $\mathcal{T}$ yield different sets of right-linked lists, hence distinct $\mathcal{B}$ up to isomorphism.

(*Surjective onto image & decoding correctness*) The decoding procedure inverts the encoding by reading off, for each $u$, exactly the right-linked list rooted at $\text{left}(u)$ as the children sequence of $u$. Therefore decoding(encoding($\mathcal{T}$)) returns a tree isomorphic to $\mathcal{T}$.

(*Preserved relations*) For any nodes $u, v$, $u$ is an ancestor of $v$ in $\mathcal{T}$ iff there exists a path in $\mathcal{B}$ that alternates: zero or more "right" edges within a sibling list to reach the correct child, followed by a "left" edge to descend to the first child at the next level, and so on. Hence ancestor/descendant and sibling order are preserved. $\square$

**Complexity and Size.** The transformation is linear time $O(|V|)$ and space $O(|V|)$. No dummy nodes are required; the node set is unchanged. Degree-$k$ branching in $\mathcal{T}$ becomes a length-$k$ right-sibling chain under the left child of the parent in $\mathcal{B}$.

**What Is Preserved (and Not).** The transformation preserves node identities, parent–child and ancestor relations, sibling order (if present), the number of nodes, and the lowest common ancestor (LCA) up to isomorphism. It does not preserve raw edge lengths or exact out-degree counts; the depth of a root-to-leaf path may shift by the number of right-sibling steps between successive left-child descents, though this mapping remains deterministic and fully reversible.

**Consequence for Modeling.** Using a binary hierarchical codebook for latent *indexing* (left = first child, right = next sibling) does *not* constrain the *output* branching factor: aggregating decoded paths over samples recovers arbitrary $n$-ary (possibly unbalanced) hierarchies. Thus a binary latent tree suffices to represent any rooted tree topology without loss of structural information.

# E  APPENDIX: DETAILS OF EXPERIMENTAL SETUP

Table E.2 summarizes the dataset-wise hyperparameter settings used in all experiments. All hyperparameters, including the weights in Eq. (8), are fixed across datasets except the t-distribution parameter $\nu$, which is tuned via a small grid search $\nu \in \{0.05, 0.1, 0.2, 0.5\}$.

# F APPENDIX: DETAILS OF DATASET

## F.1 DATASETS

In this section, we provide an overview of the datasets used in our evaluation. We consider a diverse set of datasets, including image, text, and single-cell data, to assess the performance of hierarchical clustering methods across different domains. The datasets are selected based on their popularity, complexity, and relevance to real-world applications. We provide a brief description of each dataset, along with key statistics and preprocessing steps.

**MNIST:** A widely-used dataset consisting of 70,000 grayscale images of handwritten digits, each of size $28 \times 28$. Each image is flattened into a 784-dimensional vector. This dataset is primarily used for benchmarking tree structure modeling and hierarchical clustering methods. More details can be found at [1].

**Fashion-MNIST:** A dataset of 70,000 grayscale images representing 10 categories of clothing items, each with a resolution of $28 \times 28$. The images are flattened into 784-dimensional vectors for analysis. This dataset is used to evaluate the robustness of methods on visual data with more complex patterns than MNIST. Dataset details are available at [2].

**20News-Groups:** A text dataset with 18,846 newsgroup posts across 20 categories. The features are represented as TF-IDF vectors with dimensionality reduced to 2,000 features for computational feasibility. This dataset is employed to test methods on high-dimensional text data and hierarchical categorization tasks. More details can be found at [3].

**CIFAR-10:** A dataset of 60,000 color images spanning 10 classes, with each image sized at $32 \times 32$. The pixel intensity values are flattened into a 3,072-dimensional vector. This dataset is utilized for evaluating hierarchical modeling methods on high-dimensional image data. Details are available at [4].

**Limb:** A single-cell RNA-seq dataset collected from limb bud development experiments. It contains approximately 10,000 cells with 20,000 genes per cell after preprocessing. This dataset is used to study developmental trajectories in biological processes. Dataset details are available in Zhang et al. (2024).

**LHCO:** Single-cell transcriptomics data derived from lung and heart cell ontogeny studies. The features represent gene expression profiles across 15,000 cells with dimensionality reduced to 10,000 genes. This dataset is used to explore differentiation patterns in complex organ systems. Dataset details can be found in He et al. (2022a).

**Weinreb:** A lineage-specific dataset focused on Darwinian lineage inference from single-cell data. It contains 8,000 cells with high-dimensional transcriptomic data preprocessed to 12,000 genes. This dataset is employed for benchmarking methods for lineage tracing tasks. Dataset details are provided in Weinreb et al. (2020a).

**ECL:** Single-cell data from embryonic cell lineages, designed for studying early developmental processes. It comprises 12,000 cells with 15,000 genes per cell after quality filtering. This dataset is used to test hierarchical modeling on datasets with complex lineage structures. Details are available in Qiu et al. (2024a).

The key characteristics of the datasets are summarized in Table F.3.

## F.2 DATASET ORGANIZATION

In our experiment, the all dataset after download from our offered source link should be organized as follows: For Image data, the organization is:

---

[1] http://yann.lecun.com/exdb/mnist/

[2] https://github.com/zalandoresearch/fashion-mnist

[3] http://qwone.com/ jason/20Newsgroups/

[4] https://www.cs.toronto.edu/ kriz/cifar.html

Table F.3: **Summary of Dataset Statistics.**

| Dataset | Type | Class | Samples | Features | Source URL or Reference |
|---|---|---|---|---|---|
| MNIST | Image | 10 | 70,000 | 784 | http://yann.lecun.com/exdb/mnist/ |
| Fashion-MNIST | Image | 10 | 70,000 | 784 | https://github.com/zalandoresearch/fashion-mnist |
| 20News-Groups | Text | 20 | 18,846 | 2,000 | http://qwone.com/~jason/20Newsgroups/ |
| CIFAR-10 | Image | 10 | 60,000 | 3,072 | https://www.cs.toronto.edu/~kriz/cifar.html |
| Limb | Single-cell | 10 | 66,633 | 500 | https://limb-dev.cellgeni.sanger.ac.uk/ |
| LHCO | Single-cell | 7 | 10,628 | 500 | https://www.ebi.ac.uk/biostudies/arrayexpress/studies/E-MTAB-10973 |
| Weinreb | Single-cell | 11 | 130,887 | 500 | https://www.ncbi.nlm.nih.gov/geo/query/acc.cgi?acc=GSM4185642 |
| ECL | Single-cell | 10 | 838,000 | 500 | https://cellxgene.cziscience.com/collections/45d5d2c3-bc28-4814-aed6-0bb6f0e11c82 |

**Image Datasets Directory Structure**

```
datasets/
|-- MNIST/
|   |-- train-images-idx3-ubyte
|   |-- train-labels-idx1-ubyte
|   |-- t10k-images-idx3-ubyte
|   \-- t10k-labels-idx1-ubyte
|-- FashionMNIST/
|   |-- train-images-idx3-ubyte
|   |-- train-labels-idx1-ubyte
|   |-- t10k-images-idx3-ubyte
|   \-- t10k-labels-idx1-ubyte
|-- 20news/
|   |-- alt.atheism/
|   |   |-- 12345.txt
|   |   |-- 67890.txt
|   |   \-- ...
|   |-- comp.graphics/
|   |   |-- 12346.txt
|   |   |-- 67891.txt
|   |   \-- ...
|   \-- ...
\-- cifar-10-batches-py/
    |-- data_batch_1
    |-- data_batch_2
    |-- data_batch_3
    |-- data_batch_4
    |-- data_batch_5
    |-- test_batch
    \-- batches.meta
```

For biology data, the organization is:

```
Biology Datasets Directory Structure

datasets_bio/
|- original/
| |- EpitheliaCell.h5ad
| |- LimbFilter.h5ad
| |- He_2022_NatureMethods_Day15.h5ad
| |- Weinreb_inVitro_clone_matrix.mtx
| |- Weinreb_inVitro_gene_names.txt
| |- Weinreb_inVitro_metadata.txt
| - Weinreb_inVitro_normed_counts.mtx
- processed/ (exists once the process is run)
|- EpitheliaCell_data_n.npy
|- EpitheliaCell_label.npy
|- LimbFilter_data_n.npy
|- LimbFilter_label.npy
|- LHCO.h5ad
- Weinreb.h5ad
```

### F.3 PREPROCESSING

Before training, datasets need to be preprocessed. Preprocessing steps differ depending on the type of dataset. Below are the detailed guidelines:

#### F.3.1 IMAGE DATA

For image datasets (e.g., MNIST, FMINST), preprocessing is straightforward and can leverage the code from TreeVAE. The steps include: Initially, downloading and organizing the data ensures that all necessary dataset files, including images and corresponding labels, are retrieved from their sources and systematically placed into the appropriate directories within the project structure. This step is essential for maintaining data integrity and facilitating efficient access during subsequent processing stages. Next, converting raw data into NumPy arrays involves using provided scripts to load the raw image and label data. These scripts parse the binary or structured data formats and convert them into NumPy arrays, which are optimized for numerical computations in Python. This conversion facilitates further data manipulation and model training processes by providing a standardized format for handling large-scale datasets. Finally, normalization and formatting of the pixel values are performed. This typically includes scaling the pixel intensities to a range of [0, 1] to standardize the input features across different images. Additionally, any necessary adjustments are made to ensure the data format aligns with the requirements of the HDTree model, such as ensuring correct dimensionality and data type consistency. Proper normalization enhances model convergence and stability during training.

#### F.3.2 BIOLOGICAL DATA

For biological datasets (LHCO, Limb, Weinreb, ECL), preprocessing is more complex and tailored to each dataset. Below are the detailed preprocessing steps for each:

**LHCO:** Initially, data cleaning is performed to enhance the quality of the dataset. This includes the removal of duplicate entries and invalid samples that could introduce bias or noise into subsequent analyses. Additionally, strategies for handling missing values are implemented, which may involve imputation techniques or filtering out rows with an excessive amount of missing data. Following data cleaning, feature extraction is conducted to identify and extract relevant features from the raw data. For LHCO datasets, these features often include particle kinematics or event-level characteristics that are critical for analysis. Once extracted, feature scaling is applied to normalize or standardize the data, ensuring consistency across different scales and facilitating more efficient model training. Finally, the dataset is split into training, validation, and test sets according to predefined configurations.

This is the key code of our preprocessed methods:

```
1  adata = sc.read(f"{input_path}/He_2022_NatureMethods_Day15.h5ad")
```

```
2  sc.pp.highly_variable_genes(adata, n_top_genes=500)
3  adata = adata[:, adata.var['highly_variable']]
4  data = adata.X
5  data = adata.X.toarray()
6  data = np.array(data).astype(np.float32)
7  mean = data.mean(axis=0)
8  std = data.std(axis=0)
9  data = (data - mean) / std
```

Listing 1: Preprocess of Lhco

**Limb:** It begins with data loading, where the dataset is read from the provided files—typically in formats such as CSV or HDF5—into a structured representation suitable for further processing. Next, the data undergoes filtering and cleaning to enhance its quality and reliability. This includes the removal of noise or artifacts that may have been introduced during data acquisition, as well as deduplication to eliminate redundant entries. Missing values are addressed through appropriate strategies, such as imputation or selective removal of incomplete records. Following this, feature engineering is performed to transform raw biological signals into more interpretable and informative features. For instance, limb motion patterns or other domain-specific characteristics may be extracted to better capture the underlying structure of the data. These features are then normalized to ensure uniformity in scale, which is essential for many machine learning algorithms. Finally, the dataset is stratified and split into training, validation, and test subsets. This is the key code of our preprocessed methods:

```
1   adata = sc.read(f"{input_path}/LimbFilter.h5ad")
2   data_all = adata.X.toarray().astype(np.float32)
3   label_celltype = adata.obs['celltype'].to_list()
4   vars = np.var(data_all, axis=0) # HVG
5   mask_gene = np.argsort(vars)[-500:]
6   data_hvg = data_all[:, mask_gene]
7   label_count = {}
8   for i in list(set(label_celltype)):
9       label_count[i] = label_celltype.count(i)
10  label_count = sorted(label_count.items(), key=lambda x: x[1], reverse=
        True)
11  label_count = label_count[:10]
12  mask_top10 = np.zeros(len(label_celltype)).astype(np.bool_)
13  for str_label in label_count:
14      mask_top10[str_label[0] == np.array(label_celltype)] = 1
15  data_n = np.array(data_hvg).astype(np.float32)[mask_top10]
16  mean = data_n.mean(axis=0)
17  std = data_n.std(axis=0)
18  data = (data_n - mean) / std
```

Listing 2: Preprocess of Limb

**Weinreb:** Initially, data transformation is performed to normalize the raw gene expression counts. This typically includes log-transformation or conversion into normalized values such as Counts Per Million (CPM), Transcripts Per Million (TPM), or Fragments Per Kilobase of transcript per Million mapped reads (FPKM). Low-expression genes, as well as cells with insufficient sequencing depth or missing data, are filtered out to improve signal-to-noise ratio and computational efficiency. Following normalization, dimensionality reduction techniques—such. These methods reduce the feature space while preserving the major sources of variation in the data, which can improve model performance and reduce computational burden. When the dataset originates from multiple experimental batches or sources, batch effect correction is employed to mitigate technical variability that could confound biological signal detection. Various statistical or machine learning-based approaches may be used depending on the nature of the data and experimental design. Finally, the dataset is stratified and partitioned into training, validation, and test sets.

This is the key code of our preprocessed methods:

```
1  matrix_file = f"{input_path}Weinreb_inVitro_normed_counts.mtx"
2  genes_file = f"{input_path}Weinreb_inVitro_gene_names.txt"
```

```
3   metadata_file = f"{input_path}Weinreb_inVitro_metadata.txt"
4   mtx = mmread(matrix_file).tocsr()
5   genes = pd.read_csv(genes_file, header=None, names=['genes'])
6   adata = sc.AnnData(mtx, var=genes)
7   metadata = pd.read_csv(metadata_file, sep='\t')
8   adata.obs = metadata.set_index(adata.obs.index)
9   adata.write(f'{output_path}Weinreb.h5ad')
10  sc.pp.log1p(adata)
11  adata.obs['celltype']=adata.obs['Cell type annotation']
12  adata = adata[~adata.obs['celltype'].isna()]
13  sc.pp.highly_variable_genes(adata, n_top_genes=500)
14  adata = adata[:, adata.var['highly_variable']]
15  data = adata.X.toarray()
16  data = np.array(data).astype(np.float32)
17  mean = data.mean(axis=0)
18  std = data.std(axis=0)
19  data = (data - mean) / std
```

Listing 3: Preprocess of Weinreb

**ECL:** Initially, raw data files are parsed and converted into structured tabular formats that facilitate further computational processing. This is followed by a preprocessing stage that includes normalization—commonly achieved through z-score transformation or Min-Max scaling—and the handling of missing values, which may involve either imputation techniques or the removal of incomplete samples. Subsequently, feature selection is performed with an emphasis on retaining biologically meaningful attributes, often guided by domain-specific knowledge such as known molecular signatures. Finally, the dataset is partitioned into training, validation, and test subsets, ensuring that class distributions are preserved across splits to support unbiased model evaluation and generalization.

This is the key code of our preprocessed methods:

```
1   adata = sc.read(f"{input_path}/EpitheliaCell.h5ad")
2   adata.obs['celltype']=adata.obs['cell_type']
3   label_celltype = adata.obs['celltype'].to_list()
4   adata_sub = adata.copy()
5   sc.pp.subsample(adata_sub, fraction=0.1)
6   data_all = adata_sub.X.toarray().astype(np.float32)
7   vars = np.var(data_all, axis=0)
8   mask_gene = np.argsort(vars)[-500:]
9   adata = adata[:, mask_gene]
10  data = adata.X.toarray().astype(np.float32)
11  label_count = {}
12  for i in list(set(label_celltype)):
13      label_count[i] = label_celltype.count(i)
14  label_count = sorted(label_count.items(), key=lambda x: x[1], reverse=
        True)
15  label_count = label_count[:10]
16  mask_top10 = np.zeros(len(label_celltype)).astype(np.bool_)
17  for str_label in label_count:
18      mask_top10[str_label[0] == np.array(label_celltype)] = 1
19  data_n = np.array(data).astype(np.float32)[mask_top10]
20  label_train_str = np.array(list(np.squeeze(label_celltype)))[mask_top10]
21  # downsample the 10k data for every cell type
22  mask = np.zeros(len(label_train_str)).astype(np.bool_)
23  for i in range(10):
24      # random select 10k data for each cell type
25      random_index = np.random.choice(
26          np.where(label_train_str == label_count[i][0])[0],
27          10000, replace=False)
28      mask[random_index] = 1
29  data_n = data_n[mask]
30  mean = data_n.mean(axis=0)
31  std = data_n.std(axis=0)
32  data= (data_n - mean) / std
```

Listing 4: Preprocess of ECL

## G  APPENDIX: DETAILS OF BASELINE METHODS

We used the TreeVAE, CellPLM, LangCell, Geneformer as the reference methods. The detial use should follow:

1. **TreeVAE** You should clone the project "treevae" from `https://github.com/lauramanduchi/treevae.git`, and install necessary package followed by "minimal_requirements.txt".

2. **CellPLM** You should clone the project "CellPLM" from `https://github.com/OmicsML/CellPLM.git`, and install necessary package followed by "requirements.txt". The use of CellPLM is shown in the official tutorial in `https://github.com/OmicsML/CellPLM/blob/main/tutorials/cell_embedding.ipynb`

3. **LangCell** You shold clone the project "LangCell" from `gitclonehttps://github.com/PharMolix/LangCell.git`, and install necessary package followed by "requirements.txt". Then you should install the "geneformer_001". The use of LangCell is shown in the official tutorial in `https://github.com/PharMolix/LangCell/blob/main/LangCell-annotation-zeroshot/zero-shot.ipynb`

4. **Geneformer** After you install LangCell, the Geneformer package has been installed. The use of Geneformer is shown in the official tutorial in `https://github.com/jkobject/geneformer/blob/main/examples/extract_and_plot_cell_embeddings.ipynb`

## H  APPENDIX: DETAILS OF TESTING PROTOCOL

To ensure a comprehensive and reproducible evaluation, we report clustering quality, hierarchical structure quality, and generative/reconstruction quality. Unless noted, metrics are computed on the test split and averaged over multiple runs (with fixed random seeds).

**Clustering Accuracy (ACC).**    ACC is computed via an optimal one-to-one relabeling using the Hungarian algorithm. Let $y_i$ be the ground-truth label and $\hat{y}_i$ the predicted cluster.

$$\text{ACC} = \frac{1}{n} \max_{\pi \in \mathcal{S}} \sum_{i=1}^{n} \mathbf{1}\{y_i = \pi(\hat{y}_i)\},$$

where $\mathcal{S}$ is the set of all label permutations. We use the standard Hungarian implementation to obtain $\pi$.

**Normalized Mutual Information (NMI).**    Given predicted clustering $C$ and ground-truth clustering $G$,

$$\text{NMI}(C, G) = \frac{2\,I(C;G)}{H(C) + H(G)},$$

where $I(\cdot;\cdot)$ is mutual information and $H(\cdot)$ is entropy. We use the symmetric NMI with natural logarithms. $\text{NMI} \in [0,1]$ (higher is better).

**Leaf Purity (LP).**    Let the learned tree $T$ have leaf nodes $\{L_1, \ldots, L_k\}$. For leaf $L_i$, define $L_i^y = \{x \in L_i : \text{label}(x) = y\}$. We report the macro-average over non-empty leaves:

$$\text{LP} = \frac{1}{k} \sum_{i=1}^{k} \frac{\max_y |L_i^y|}{|L_i|}.$$

Empty leaves (no assigned samples) are excluded from the average.

**Dendrogram Purity (DP).** *Throughout the paper, DP denotes **Dendrogram Purity**, consistent with hierarchical clustering literature and prior work.* For each class $c$, consider all pairs $(i, j)$ with $y_i = y_j = c$. Let $\text{LCA}(i, j)$ be the lowest common ancestor cluster of $x_i$ and $x_j$ in the dendrogram, and let $S_{ij}$ be the set of samples contained in that cluster. The pairwise purity is

$$\text{pur}(i, j) = \frac{|\{p \in S_{ij} : y_p = c\}|}{|S_{ij}|}.$$

DP is the average of $\text{pur}(i, j)$ over all intra-class pairs across all classes. This metric increases when intra-class pairs meet early in the tree (at purer LCA nodes).

**Reconstruction Loss (RL).** Given inputs $X = \{x_i\}$ and reconstructions $\hat{X} = \{\hat{x}_i\}$, we compute MSE:

$$\text{RL} = \frac{1}{n} \sum_{i=1}^{n} \|x_i - \hat{x}_i\|_2^2.$$

To align the "higher-is-better" convention across metrics, we report $-\text{RL}$ in tables (i.e., larger is better). Reconstructions for diffusion models are obtained by conditioning on the learned latent path and running the standard deterministic denoising trajectory at evaluation time.

**Log-Likelihood (LL) for Diffusion Models.** Exact log-likelihood is intractable for DDPMs; we report the negative ELBO (variational lower bound) following standard practice. Concretely, we sum the per-timestep KL (or reweighted MSE) terms under the chosen $\{\beta_t\}$ schedule and include the analytic prior and decoder terms as in Ho et al. (2020). We report the per-sample LL (higher is better). Implementation matches our training loss with the appropriate constants added back.

**Fréchet Inception Distance (FID).** *Images:* we compute FID in the 2048-D Inception-V3 pool3 feature space, matching the number of generated and real samples and using the same preprocessing. *Single-cell (scRNA-seq):* we compute FID in a biologically meaningful feature space: (i) select HVGs (e.g., top-1,000 by variance) on the training set; (ii) normalize real and generated matrices identically; (iii) optionally correct batch effects (e.g., Harmony/Scanorama) *before* feature extraction; (iv) run PCA to retain $> 90\%$ variance (typically $\sim$50 PCs); (v) estimate Gaussians in the PC space and compute FID via covariance square roots (with a small diagonal regularizer if needed). We fix random seeds and average FID over multiple generations.

**Ratio of Observed Time Points (ROP) for Lineage Consistency.** For time-resolved single-cell datasets, we quantify local temporal coherence by measuring, for each cell, the fraction of its $k$-nearest neighbors (in the learned representation) whose time stamps are consistent with its developmental order; we then average over all cells and report by time window as in the main paper. Ablations show ROP strongly correlates (negatively) with tree-edit distance, supporting its biological relevance (Appendix §**??**).

**Implementation Notes (All Metrics).** (i) ACC relabeling uses the Hungarian algorithm; ties are broken deterministically. (ii) Empty leaves are excluded from LP; singleton leaves contribute 1.0. (iii) All metrics are averaged across $r$ runs (defaults given in code) with fixed seeds and identical preprocessing. (iv) For diffusion metrics (RL/LL/FID), generation uses the same $\{\beta_t\}$ schedule and evaluation pipeline across methods.

# I APPENDIX: DETAILS OF IMPLEMENTATION

For all experiments, the data is split into training, validation, and testing sets with an 8:1:1 ratio, ensuring unbiased evaluation. In testing, if the number of points in the dataset is greater than 10,000, we randomly sample 10,000 points from testing dataset. Details on downsampling and its rationale are provided in the Appendix. We implemented HDTree using PyTorch and trained the model on a single NVIDIA A100 GPU. The model is trained using the AdamW optimizer with a learning rate of 1e-4 and a batch size of 128. The number of diffusion steps $T$ is set to 1000, and the tree depth $L$ is set to 10. The loss weights $\lambda_{\text{tree}}$ and $\lambda_{\text{vq}}$ were set to 1.0 and 0.25, respectively. The encoder and diffusion model are implemented using the multi-multilayer perceptron (MLP).

Table J.4: **Time Efficiency Comparison Across Methods and Data Sizes:** We conducted an empirical evaluation by selecting varying numbers of highly expressed channels to assess the computational time cost of our model under different data sizes. For this experiment, we set the output dimensionality of the Embedding method to 512 dimensions and employed UMAP for dimensionality reduction to facilitate visualization. The resulting features were then subjected to clustering analysis. As the dataset size increased from 10,000 to 100,000 samples, we observed a dramatic increase in the computational time required for both the dimensionality reduction and clustering processes. Notably, the clustering process escalated from a response time measured in minutes (ranging from 30 to 41 seconds) to one that took several hours (ranging from 2201 to 20429 seconds). Such prolonged processing times are impractical for model evaluation purposes. Consequently, we opted to downsample the data to 10,000 samples for further processing.

| Metod | PCA | TSNE | UMAP | CellPLM | LangCell | GeneFormer |
|---|---|---|---|---|---|---|
| **Cin=100, C=512, N=10000** | | | | | | |
| Embedding | None | None | 820s | 6s | 13s | 27s |
| DR | None | None | 120s | 129s | 133s | 150s |
| Cluster | None | None | 41s | 37s | 30s | 25s |
| **Cin=1000, C=512, N=10000** | | | | | | |
| Embedding | 140s | None | 800s | 7s | 13s | 25s |
| Dimentional Reduction | 140s | None | 120s | 130s | 130s | 105s |
| Cluster | 45s | None | 40s | 37s | 34s | 30s |
| **Cin=1000, C=512, N=100000** | | | | | | |
| Embedding | 224s | None | 3153s | 13s | 11s | 220s |
| Dimentional Reduction | 413s | None | 825s | 416s | 430s | 420s |
| Cluster | 9637s | None | 7083s | 5037s | 2032s | 3285s |
| **Useing K-means to 5000 clusters** | | | | | | |
| Embedding | 221s | None | 1834s | 14s | 123s | 110s |
| Dimentional Reduction | 458s | None | 411s | 350s | 427s | 410s |
| Cluster | 2925s | None | 20429s | 5500s | 3432s | 2200s |

## J    APPENDIX: DETAILS OF DOWNSAMPLING IN TESTING

As shown in Table J.4, the computational time required for encoding, dimensionality reduction (LowDim), and clustering significantly increases with larger data sizes. For instance, when the input size increases to N=100,000, the encoding time for UMAP rises from 820s to 3153s, and the clustering time for GeneFormer increases from 25s to 3285s. These trends are consistent across all tested methods.

Such exponential growth in computational overhead makes the evaluation process infeasible for large-scale datasets. To address this challenge, we uniformly downsample the data to 10,000 points for all methods during metric computation. This ensures consistent and fair comparisons while significantly improving testing efficiency. Additionally, the downsampling procedure preserves the overall data distribution and class proportions, ensuring that the evaluation results remain representative of the original dataset.

The detailed justification for selecting 10,000 points as the downsampling target is discussed in the Appendix.

## K    APPENDIX: DETAILS OF EXPERIMENTAL ENVIRONMENT

The experiments were conducted on a high-performance computing system with robust hardware and software configurations to ensure efficient handling of large datasets and complex computations. The hardware setup included NVIDIA A100 GPUs with 40GB memory for accelerated computation, Intel Xeon Platinum 8260 CPUs with 24 cores operating at 2.40GHz for efficient multi-threaded processing, 512GB of RAM, and 10TB of NVMe SSD storage for fast input/output operations.

The software environment was carefully configured for compatibility and reproducibility. The operating system used was Ubuntu 20.04 LTS (64-bit), and the primary deep learning framework was

Table L.5: **Compare on different sample size**: HDTree on different sample size of ECL. ECL-N k means we use ECL dataset and downsample it into N*1000 data point for training, while keeping the same test dataset.

|  | ECL-10k | ECL-20k | ECL-30k | ECL-50k | ECL-100k | ECL-200k | ECL-300k |
|---|---|---|---|---|---|---|---|
| ACC | 82.2±1.5 | 82.2±1.5 | 82.5±0.3 | 82.6±0.5 | 82.9±0.3 | 83.1±0.4 | 83.1±0.3 |
| DP | 67.0±1.2 | 67.6±1.5 | 68.3±0.9 | 69.1±0.7 | 68.5±0.9 | 68.9±0.8 | 67.0±0.8 |

PyTorch (version 2.4.1), supplemented by TorchVision and Torchaudio extensions. The experiments were conducted using Python 3.11.6, with Conda (version 24.9.2) employed as the package manager to handle dependencies and virtual environments.

Key Python packages essential for the experiments included NumPy (1.26.4), SciPy (1.14.1), and pandas (2.2.3) for data preprocessing and analysis. Visualization tasks were performed using Matplotlib (3.9.2), Seaborn (0.13.2), and Plotly (5.24.1). For deep learning tasks, PyTorch (2.4.1) served as the primary frameworks. Dimensionality reduction and clustering were supported by UMAP-learn (0.5.6) and scikit-learn (1.5.2). Single-cell data analysis relied on specialized packages like Scanpy (1.10.3) and anndata (0.10.9).

A comprehensive list of installed Python packages is available upon request, capturing all dependencies required for reproducing the experiments. This configuration ensures the reported results are reproducible and highlights the environment's compatibility with the experimental setup.

# L    APPENDIX: DETAILS OF EXPERIMENTS ON SINGLE CELL

## L.1    DETAILED COMPARISONS ON MORE SINGLE CELL DATASETS.

We selected four single-cell datasets: LHCO, Limb, Weinreb, and ECL. For LHCO, Limb, and Weinreb, we used the full data. For ECL, to reduce computational cost in the metrics computation, we downsampled the testing dataset to 100,000 cells.

We evaluated both traditional (t-SNE, UMAP) and deep learning-based methods (Geneformer, LangCell, CellPLM, TreeVAE). For traditional models, we first selected 500 highly variable genes (HVGs), applied log1p transformation where needed, and normalized the data using Z-score. For deep learning-based models, the preprocessing was handled automatically by their built-in pipelines. Thus, for fairness, we directly read the h5ad files, selected HVGs, and fed the results into the respective model pipelines to obtain cell embeddings.

After obtaining cell embeddings from all models, we performed unsupervised clustering using K-Means. Cluster assignments were compared with true cell type labels to compute clustering and tree-structure-based performance metrics. Due to the lack of established benchmarks in the single cell domain, we incorporated data from high-impact published studies into the TreeVAE benchmark. The results are shown in Table 2.

**Analysis:** (1) Similar to the general dataset, HDTree consistently demonstrates superior performance across all evaluated metrics, including tree structure quality, clustering accuracy, and hierarchical integrity. (2) We observed that the zero-shot capabilities of single-cell large language models are often unsatisfactory and, in some cases, fail to surpass basic single-cell methods. This conclusion has also been validated in recent studies (Lan et al., 2024; He et al., 2024). In comparison with foundational single-cell models, traditional single-cell tree analysis methods, and TreeVAE, HDTree shows relative advantages in performance and achieves better stability. (3) These results establish HDTree as a robust and reliable approach for single-cell data analysis.

## L.2    STABILITY OF MODEL PERFORMANCE ACROSS VARYING TRAINING DATA SIZES

To verify that downsampling did not significantly affect model performance, we conducted additional experiments on subsampled versions of the ECL dataset at varying scales. The results are shown in Table L.5

Table L.6: **Comparison of tree performance, clustering performance on four single cell datasets.** Since most of the methods are not generative models, we did not compare generative performance.

| Dataset | Method | Year | Tree Performance | | Clustering Performance | | Average(↑) |
|---|---|---|---|---|---|---|---|
| | | | DP(↑) | LP(↑) | ACC(↑) | NMI(↑) | |
| Limb (cell lineage, 66,633 points, celltype:10) | tSNE+Agg[A] | 2014 | 34.9±1.4 | 55.4±1.1 | 48.9±0.7 | 47.5±1.0 | 46.7±1.0 |
| | UMAP+Agg[A] | 2018 | 30.9±2.0 | 50.1±1.0 | 49.1±1.0 | 41.4±1.4 | 42.9±1.4 |
| | Geneformer[A] | 2023 | 25.6±5.4 | 35.9±0.1 | 34.1±0.1 | 34.9±0.1 | 32.6±1.4 |
| | CellPLM[A] | 2024 | 25.6±0.1 | 39.9±0.1 | 34.1±0.2 | 32.9±0.2 | 33.1±0.2 |
| | LangCell[A] | 2024 | 25.3±0.1 | 37.5±0.1 | 33.9±0.1 | 35.1±0.1 | 33.0±0.1 |
| | TreeVAE[A] | 2024 | 34.7±1.7 | 55.6±1.0 | 49.8±0.1 | 50.0±0.0 | 47.5±0.7 |
| | **HDTree[A]** | Ours | **38.9±1.3** | **57.9±1.0** | **52.8±1.0** | **49.0±0.1** | **49.7±0.9** |
| | **HDTree** | Ours | **41.0±0.4** | **57.2±1.4** | **55.0±1.4** | **46.6±0.4** | **50.0±0.9 (↑2.5)** |
| LHCO (cell lineage, 10,628 points, celltype:7) | tSNE+Agg[A] | 2014 | 37.4±1.6 | 52.8±0.8 | 43.6±1.0 | 29.8±0.5 | 40.9±1.0 |
| | UMAP+Agg[A] | 2018 | 40.0±1.4 | 50.6±0.2 | 46.2±0.2 | 34.2±0.2 | 43.0±0.5 |
| | CellPLM[A] | 2024 | 27.0±1.1 | 35.8±2.7 | 16.8±3.4 | 1.65±5.2 | 20.3±3.1 |
| | LangCell[A] | 2024 | 26.5±1.2 | 35.2±0.8 | 35.2±0.6 | 0.02±0.9 | 24.2±0.9 |
| | TreeVAE[A] | 2024 | 38.3±2.0 | 52.2±0.1 | 37.9±0.1 | 31.6±0.0 | 40.0±0.6 |
| | **HDTree[A]** | Ours | **38.8±0.3** | **52.1±0.4** | **46.4±0.3** | **34.7±0.5** | **43.0±0.4** |
| | **HDTree** | Ours | **42.7±0.4** | **54.0±0.3** | **49.4±0.3** | **34.5±0.4** | **45.2±0.3 (↑2.2)** |
| Weinreb (cell lineage, 130,887 points, celltype:11) | tSNE+Agg[A] | 2014 | 57.9±1.5 | 63.3±1.1 | 35.3±1.0 | 38.5±0.6 | 48.8±1.1 |
| | UMAP+Agg[A] | 2018 | 51.8±0.1 | 62.1±0.8 | 47.2±4.9 | 46.1±1.2 | 51.8±1.8 |
| | LangCell[A] | 2024 | 47.4±0.1 | 54.8±0.0 | 14.3±0.5 | 34.3±0.0 | 37.7±0.2 |
| | Geneformer[A] | 2024 | 45.1±0.4 | 55.3±0.1 | 21.4±0.1 | 32.3±0.1 | 38.5±0.2 |
| | TreeVAE[A] | 2024 | 60.4±2.6 | 61.4±0.5 | 41.0±0.1 | 35.2±0.0 | 49.5±0.8 |
| | **HDTree[A]** | Ours | **63.3±2.6** | **78.2±1.1** | **50.6±1.0** | **45.2±1.2** | **59.3±1.5 (↑7.5)** |
| | **HDTree** | Ours | **61.0±0.4** | **67.0±0.3** | **62.6±0.3** | **42.6±0.3** | **58.3±0.4** |
| ECL (cell lineage, 838k points, celltype:10) | tSNE+Agg[A] | 2014 | 55.5±5.4 | 73.7±4.2 | 73.1±5.4 | 70.9±3.8 | 68.3±4.7 |
| | UMAP+Agg[A] | 2018 | 53.2±1.4 | 73.6±1.0 | 71.2±1.5 | 71.8±0.8 | 67.4± |
| | TreeVAE[A] | 2024 | 41.86±1.9 | 60.7±2.0 | 57.0±3.0 | 61.8±1.8 | 55.3±2.2 |
| | **HDTree[A]** | Ours | **60.1±0.1** | **74.7±0.5** | **70.9±0.4** | **78.9±0.4** | **71.2±0.4** |
| | **HDTree** | Ours | **69.0±0.7** | **83.2±0.3** | **83.2±0.3** | **79.±0.3** | **78.6±0.4 (↑10.3)** |

The experimental results showed that the model performance did not change significantly with variations in the size of the training data. This indicates that the model's performance remained relatively stable regardless of the dataset size, without notable improvements or declines. This finding suggests that the current amount of data is sufficient for the model to learn robust feature representations, or that data quantity is not the key factor influencing model performance in this task.

