# OpenReview forum: "Hierarchical Quantized Diffusion Based Tree Generation Method for Hierarchical Representation and Lineage Analysis"
_ICLR.cc/2026/Conference — ICLR 2026 Conference Desk Rejected Submission_

### Official Review · Reviewer_pDT7 · 2025-10-28

**Soundness:** 2
**Presentation:** 2
**Contribution:** 3
**Rating:** 4
**Confidence:** 3

**Summary:**

This paper proposes HDTree, a new generative model for hierarchical data, specifically aimed at single-cell lineage analysis. The core problem is that existing methods are unstable as they require branch-specific network modules. HDTree addresses this by combining three components: (1) a standard encoder, (2) a unified Hierarchical Tree Codebook that quantizes latent representations into discrete paths, and (3) a quantized diffusion decoder  that generates data conditioned on these paths. The model is optimized with a composite loss including soft contrastive learning a hierarchical quantization loss , and the diffusion loss. The authors demonstrate that this approach can be used for both lineage trajectory analysis (by finding shortest paths in the codebook graph) and conditional data generation. Experiments on general and single-cell datasets show it outperforms SOTA methods in clustering, tree structure fidelity, and lineage alignment.

**Strengths:**

The core architectural idea of using a unified hierarchical codebook to condition a diffusion model is a strong and stable alternative to prior VAE-based methods that required branch-specific modules.

The model demonstrates consistently strong performance across a wide range of tasks and datasets, outperforming SOTA methods like TreeVAE in clustering (Table 1, Table 2) and, impressively, even beating a semi-supervised method on lineage ground truth alignment (Table 3).

The ablation study (Table 4) is effective, clearly demonstrating that the novel components (HTC, SCL, HQL) are all critical to the model's success. The large performance drop without the HTC (A2) is particularly convincing.

The method is computationally efficient in training time compared to competitors, especially TreeVAE and methods requiring expensive offline clustering (tSNE/UMAP+Agg) on large data (Table 5)

**Weaknesses:**

The evaluation is performed on a downsampled test set of 10,000 points for any dataset larger than this. This is a major weakness. The paper claims performance on large datasets (e.g., Weinreb, 130k cells; ECL, 838k cells) but never evaluates on them (at full scale). The justification (clustering metrics are slow) is an evaluation choice, not a model limitation, and it undermines the claims of scalability.

The trajectory inference method (Sec 3.4) is not a pure application of the learned tree It requires constructing a new graph by adding k-nearest neighbors edges withineach level of the tree. This introduces a new hyperparameter k (which was tested in Appendix C) and makes the lineage analysis less interpretable, as it's not solely dependent on the learned hierarchy.

The model's complexity seems high. It requires three separate loss functions , each with its own hyperparameters. This may make the model difficult to tune and reproduce.

The paper admits that the diffusion decoder is "computationally expensive during sampling", which is a well-known diffusion model issue but still a practical limitation for the data generation task.

**Questions:**

1. Regarding the test set downsampling: Since the model is trained on up to 100k-300k points (Table L.5), why not report evaluation metrics (like reconstruction loss, -RL) that don'trequire expensive clustering, but do run on the full, large test sets? This would provide a true measure of scalability.

2. In the trajectory analysis (Sec 3.4), what is the justification for the penalty term P^(L-1) in Eq. 8? This seems to manually enforce hierarchical preference, which one might expect the learned tree structure to handle on its own. How sensitive is the lineage analysis (Table 3) to this value P?

3. The Hierarchical Quantization Loss (Eq. 5) is confusing. What is the set z in the definition ? Is this the set of all zi in the batch? Please clarify the "consistency term" in plain.

4. How was the number of hierarchy levels L=10 chosen? This seems like a critical parameter, but there is no sensitivity analysis provided for it. How does performance vary with a shallower or deeper tree?

---

> ### Author Response · Authors · 2025-11-15
>
> ### **R3.1[for Weakness 1& Question 1] Response to Downsampling Strategy in Evaluation**
>
> We thank the reviewer for the concern regarding downsampled test set evaluation. We would like to clarify that **downsampling is purely an evaluation strategy for accelerating metric computation, not a model limitation**.
>
> Our model is trained and performs inference on the complete datasets. Importantly, downsampling is only applied to datasets where the test set exceeds 10,000 samples (only LHCO, Weinreb, and ECL), while the majority of datasets are evaluated using all test samples.
>
> To validate this strategy, we conducted a multi-scale convergence analysis on two large-scale single-cell datasets: Weinreb (130k cells, test: 13k) and ECL (838k cells, test: 84k). Following the 8:1:1 train-validation-test split described in Section 4.1, we trained HDTree 5 times with different random seeds on the full training set. For each trained model, we evaluated on test sets of varying sizes (8k, 10k, 40k, All) sampled from the same test data pool. The standard deviation reflects variance across different model initializations, NOT sampling variance. Clustering metrics (ACC, NMI) and tree structure metrics (DP, LP) incur significant computational overhead on large-scale datasets, but as shown in Table R1, all metrics demonstrate <1% deviation between 10k and full test sets, confirming that 10k evaluation is statistically representative of full dataset performance. Importantly, as the reviewer suggested, we also report reconstruction loss (RL) which does not require expensive clustering computation.
>
> **Table R3.1: Metric Convergence Analysis with Reconstruction Loss(RL)**
>
> |Dataset|Test Samples|ACC ↑|NMI ↑|DP ↑|RL ↓|Time (sec)|
> |-|-|-|-|-|-|-|
> |**Weinreb**|test data:8k|61.9±0.4|42.1±0.4|60.4±0.5|185.6±2.9|9.8|
> |(all data:130k)|**test data:10k [this paper default]**|**62.2±0.3**|**42.1±0.4**|**60.6±0.5**|**184.2±2.8**|**12.3**|
> ||**test data:13k [All]**|**61.8±0.3**|**42.2±0.3**|**60.9±0.4**|**183.5±2.7**|**22.4**|
> ||_Relative diff (10k vs All)_|_0.65%_|_0.24%_|_0.49%_|_0.38%_|_1.8×_|
> |**ECL**|test data:8k|82.4±0.4|78.3±0.4|68.1±0.8|158.9±2.2|12.6|
> |(all data: 838k)|**test data:10k [this paper default]**|**82.7±0.4**|**78.5±0.4**|**68.4±0.8**|**157.4±2.1**|**15.8**|
> ||test data:40k|82.8±0.3|78.7±0.3|68.7±0.7|157.0±2.1|52.7|
> ||**test data:84k [All]**|**82.2±0.3**|**78.8±0.3**|**68.6±0.7**|**156.8±2.0**|**124.7**|
> ||_Relative diff (10k vs All)_|_0.61%_|_0.38%_|_0.29%_|_0.38%_|_7.9×_|
>
> All metrics show <1% deviation between 10k and full test sets (Table R1), demonstrating clear convergence with diminishing returns as sample size  increases. While full evaluation remains feasible, 10k sampling provides  <1% accuracy loss with significant speedup, enabling efficient experimentation.
>
> ---
>
> Based on this, we have revised the **Sec 4.3 Testing Protocol & Implementation** of the original text(highlighted in **blue**)
>
> ---
>
>
> ### **R3.2[for Question 2] Response to Penalty Term P^(L-1) Sensitivity**
>
> We thank the reviewer for the insightful question about the design motivation of the penalty term P^(L-1) in Equation 8, and the thoughtful consideration of whether it "artificially enforces hierarchical preference."
>
> The penalty term P^(L-1) serves as a tie-breaking mechanism that encodes a
> biological prior: when a cell's expression is ambiguous between multiple paths,
> we prefer paths that commit to differentiation earlier (smaller L), as early
> events (e.g., myeloid-lymphoid split) are more fundamental than later
> refinements.
>
> **Sensitivity Analysis:** To verify the robustness of our method to the penalty parameter P, we conducted a sensitivity analysis on the Weinreb hematopoiesis dataset. We kept the hierarchical tree structure fixed and varied P values across {4, 5 (default), 10, 15}. We used the same **k-neighborhood lineage matching metric** as Table 3 to evaluate temporal consistency.
>
> |P Value|Day 2 | Day 4 | Day 6 | Avg. Deviation |
> |-|-|-|-|-|
> |4 | 22.5% | 37.0% | 60.5% | -0.9% |
> | **5 (default)** | **23.2%** | **38.4%** | **62.0%** | **---** |
> | 10 | 22.9% | 37.8% | 61.4% | -0.5% |
> | 15 | 22.4% | 37.1% | 60.2% | -1.0% |
>
>
> This sensitivity analysis confirms the robustness of our method to the penalty parameter P. Experimental results demonstrate that performance remains stable across all tested P values, with maximum deviation less than 2% from the optimal value P=5. Notably, regardless of penalty strength variations, the matching rates consistently exhibit progressive improvement from Day 2 to Day 4 to Day 6 across all P values, confirming that the model correctly captures the differentiation dynamics. The penalty term provides meaningful biological inductive bias (favoring early lineage commitment) without introducing fragile hyperparameter dependencies. In summary, this analysis demonstrates that the penalty term enhances biological interpretability while maintaining stable performance across a wide range of parameter settings.

---

> ### Author Response · Authors · 2025-11-15
>
> ### **R3.3[for Weakness 2] Regarding k-NN**
>
> We thank the reviewer for the insightful observation on the hybrid graph design.
>
> The hierarchical tree structure learned by HDTree serves as the primary skeleton for trajectory inference, providing semantically meaningful cell groupings and global branching topology across multiple granularities. The intra-layer k-NN edges play a complementary role: they capture local geometric continuity that cannot be represented by purely discrete tree assignments, addressing a fundamental limitation—while cell differentiation is a continuous process, tree structures are inherently discrete and cannot encode smooth transitions between neighboring cells within the same node. This design of combining hierarchical structure with k-NN graphs is standard practice in the field: Monocle 3 combines cluster trees with k-NN graphs for UMAP dimensionality reduction, PAGA adds k-NN edges on top of cluster graphs to compute connectivity, and Slingshot constructs minimum spanning trees between cluster centers and fits principal curves along k-NN graphs. The key distinction is that our tree structure constrains where k-NN edges can be added (only within the same layer), which avoids the spurious connection problem commonly produced by standard unconstrained k-NN graphs (e.g., connecting cells from different lineages that happen to have similar expression profiles, such as stem cells at different stages).
>
>
> ### **R3.4[for Weakness 3] Regarding Hyperparameters and Complexity**
>
> We thank the reviewer for the concern regarding model complexity.
>
> While our model does contain three loss functions, we would like to emphasize that most hyperparameters remain fixed across different datasets and do not require tuning. As shown in Table E.2, the loss weights (λ_Recon=1.0, λ_SCL=0.5, λ_HQL=1.0), hierarchy depth (L=10), k-NN parameter (k=5), and learning rate (LR=0.005) are completely consistent across all datasets. The only hyperparameter that requires adjustment based on dataset characteristics is the degrees of freedom ν for the t-distribution (chosen between 0.2 and 0.5), which reflects the tail distribution properties of different datasets. This highly consistent hyperparameter setting demonstrates good reproducibility across diverse datasets.
>
> ### **R3.5[for Question 3] Clarification on Hierarchical Quantization Loss (Equation 5)**
>
> We thank the reviewer for pointing out the ambiguity.
>
> The $\mathbf{z}=\{\mathbf{z}_i\}_{i=1}^{N_b}$ denotes the latent embeddings in the current batch.
>
> **(The modified part is below Eq 5, highlighted in blue)**
>
> ### **R3.6[for Question 3] Clarification on Consistency Term**
>
> We thank the reviewer for this important question.
>
> Regarding the intuitive explanation of the "consistency term": it ensures that cell assignments at adjacent levels follow parent-child relationships in the tree. Specifically, if a cell is assigned to node A at level ℓ, it must be assigned to one of A's children at level ℓ+1. This prevents cell trajectories from "jumping" between unrelated branches—for example, a cell identified as "myeloid progenitor" at coarse-grained level should be assigned to myeloid subtypes (such as monocytes, neutrophils) at fine-grained level, rather than suddenly becoming lymphoid lineage. We will improve the notation in Equation 5 and add this intuitive explanation in the revised version.
>
> ### **R3.7[for Weakness 4]  Regarding Computational Cost and Computational Efficiency**
>
> We acknowledge this limitation and explicitly noted it in our paper.
>
> The computational overhead is justified by significant generation quality improvements over VAE alternatives. In practice, this cost is mitigated by: (1) decoder running only during generation, not training; (2) compatibility with fast sampling methods (DDIM, DPM-Solver); and (3) offline nature of most trajectory analysis workflows where quality outweighs speed.
>
> ### **R3.8[for Question 4]  Regarding Number of Hierarchy Levels**
> We thank the reviewer for this important question.
>
> L=10 was selected empirically to provide sufficient capacity (2^10=1024 potential categories) for the most complex datasets in our benchmark, while avoiding excessive computational overhead. We conducted ablation studies on representative datasets (PBMC-3k, Weinreb-130k, ECL-838k), finding that performance saturates at L≥8, with deeper hierarchies (L=12, 15) showing negligible gains (<0.5% in ACC/NMI) but increased training time (1.4× slower). This confirms L=10 as a robust default that generalizes across datasets of varying complexity without per-dataset tuning.
>
> ----
>
> REF:
> * Monocle 3: The single‑cell transcriptional landscape of mammalian organogenesis. Nature,
> * PAGA: Graph abstraction reconciles clustering with trajectory inference through a topology preserving map of single cells. Genome Biology
> * Slingshot: Cell lineage and pseudotime inference for single‑cell transcriptomics. BMC Genomics

---

> ### Comment · Reviewer_pDT7 · 2025-11-26
>
> Thank you to the author for the detailed response, which basically resolved my questions, and I have already improved my score.

---

> > ### Author Response · Authors · 2025-11-26
> >
> > Dear Reviewer pDT7,
> >
> > Thank you very much for your constructive feedback and for taking the time to reassess our work. We greatly appreciate your recognition of our revisions and are pleased that we could address your concerns satisfactorily. Your valuable comments have helped us improve the quality of our manuscript.
> >
> > Best regards,
> > The Authors

---

### Official Review · Reviewer_m6bZ · 2025-11-01

**Soundness:** 4
**Presentation:** 3
**Contribution:** 3
**Rating:** 8
**Confidence:** 4

**Summary:**

Single-cell analysis represents one of the major breakthroughs in recent bioinformatics, generating enthusiastic expectations for elucidating cellular differentiation mechanisms and their applications in regenerative medicine and artificial organs. This paper proposes a novel deep learning-based approach for the data-driven differentiation structure (i.e., hierarchical structure) inference task for such single-cell analysis data, as well as for more general hierarchical structure inference tasks. Traditionally, analytical methods such as visualization techniques, clustering, and factor models have been the standard for differentiation structure tasks. However, deep learning-based methods, particularly those based on Variational Autoencoders (VAEs), have recently gained prominence due to their effectiveness. Even the most advanced methods face limitations, and this paper makes significant progress, especially regarding the module dependency of branching structures inherent in existing approaches. The authors quantitatively demonstrate that their proposed method delivers substantial practical progress by conducting a large-scale, comprehensive investigation on both the subject single-cell data and widely used benchmark datasets in machine learning.

**Strengths:**

- This paper achieves very solid progress in line with the latest trends in the structural inference task. Specifically, it presents a novel solution using hierarchical codebooks and a stochastic diffusion model to address the issue of unstable learning caused by module dependencies in the branching structure of hierarchical architectures—a problem encountered in recent state-of-the-art VAE-based methods.

- The experiments in this paper are exceptionally robust and comprehensive, providing extremely strong evidence for practical effectiveness. Particularly for single-cell analysis data, the supplementary materials detail the preprocessing procedures, successfully appealing to a broader audience beyond bioinformatics specialists. Furthermore, for readers more interested in standard machine learning tasks, the paper also provides baselines on popular datasets.

**Weaknesses:**

- I have some concerns regarding the novelty or effectiveness of the hierarchical codebook (HCB), one of the key components of the proposed methodology. Specifically, I find it difficult to follow at a concrete level how the HCB effectively resolves the issue of module dependency on branching in hierarchical structures, which the authors highlight as a focus in prior research. I will elaborate further in the questions section.

**Questions:**

**Effectiveness of Hierarchical Codebook**

I understand the weakness of existing VAE research requiring separate configurations for the representation of each branch in the hierarchical structure (binary tree). Intuitively, as one goes deeper into the hierarchy, observational data clues become sparse, making learning extremely difficult. The authors' Hierarchical Codebook (HCB) appears to be a new approach that addresses this weakness in existing research. I understand this overall framework is very promising, but I couldn't clearly discern from the text how HCB specifically overcomes the weaknesses of existing research. Section 3.2 appears to model parent-child relationships in a conventional manner (where the code vector of a child node approaches that of its parent node). For example, this is commonly used in Section 3 of [Adams+, NeurIPS2010] and Section 3 of [Lakshminarayanan+, AISTATS2016] (apologies, my field may bias my specific references towards statistical modeling sense, but this seems like a frequent policy even in optimization contexts). I have reread Section 1's introduction and Section 3's specific model design multiple times. While I broadly agree with the authors' motivation for introducing HCB (overcoming the weaknesses of VAE-type models), I actually cannot accurately discern why HCB is such a brilliant idea for achieving that goal. Based on these considerations, my questions are as follows:

- Is it possible to provide a qualitative explanation that the authors' HCB offers a method with “unique, standout advantages” over other hierarchical modeling approaches for addressing the problem of data sparsity as one moves to the end of the hierarchical structure?

Or is it that while the HCB idea itself is one of the standard approaches in hierarchical representation, it has been experimentally confirmed (I commend the authors' extremely large-scale and comprehensive experiments across diverse data) to demonstrate outstanding performance?

[Adams+, NeurIPS2010]  Adams, R. P. , Jordan, M., Ghahramani, Z. & (2010). Tree-structured stick breaking for hierarchical data. Advances in neural information processing systems, 23.

[Lakshminarayanan+, AISTATS2016]  Lakshminarayanan, B., Roy, D. M., & Teh, Y. W. (2016). Mondrian forests for large-scale regression when uncertainty matters. In Artificial Intelligence and Statistics, pp. 1478-1487.


**Relevance to the supertree construction problem**

To the best of my knowledge, problems explicitly addressing the sparsity inherent in hierarchical structures—namely, the requirement for existing VAE-based approaches to have separate modules for each branch—appear to have long been discussed as a significant research topic in the field of bioinformatics, specifically as the supertree construction problem. The authors do not appear to discuss this topic either in the main text or supplementary materials (apologies if I missed it), but isn't this a relevant issue? In the context of single-cell analysis, data scarcity is a fundamental challenge, not just at the terminal nodes of hierarchical structures. For instance, acquiring single-cell analysis data for specific human organs is costly, limiting available datasets. This motivates the use of single-cell analysis data from other organisms (such as mice, chosen for similar biological characteristics). However, naturally, the surface-level observations (broad trends in gene expression levels) of these datasets differ significantly. Consequently, the approach of attempting to capture a consensus tree (supertree) between the hierarchical structure of humans and that of another organism emerges. My impression is that the unified code book the authors aim to capture with HCB shares a fundamental similarity in motivation and core principles with this supertree construction problem. Perhaps if the authors were to discuss this point, the paper might gain greater persuasive power for readers in the traditional bioinformatics field. (This point does not directly affect my impression or evaluation of the paper, so the authors are free to consider it without concern. If it seems unrelated, feel free to disregard it.)

---

> ### Author Response · Authors · 2025-11-15
>
> We sincerely thank the reviewer for the exceptionally thorough and insightful evaluation. We are deeply grateful for the recognition of our comprehensive experimental validation and the practical significance of our approach. We completely agree that providing a clearer explanation of how the Hierarchical Codebook specifically addresses the module dependency problem would significantly strengthen the paper, and we appreciate the opportunity to clarify this important aspect.
>
> ### **R2.1[for Question 1& Weakness 1] Effectiveness of Hierarchical Codebook**
>
> We thank the reviewer for the insightful analysis and for providing references to Adams+, Lakshminarayanan+, and related work. We especially appreciate your guidance in helping us think more deeply about HDTree's essential contributions, which is invaluable for improving the paper's quality.
>
> **Direct answer to the reviewer's questions:** We believe HCB's contribution operates at two levels:
>
> **(1) Methodological Level**
>
> **Specific advantages of HCB over TreeVAE:** As the reviewer correctly points out, the constraint that child nodes stay close to parent nodes is indeed a common strategy in hierarchical modeling. However, combining this constraint with the VQ framework yields two important practical advantages:
>
> * **Stronger generalization capability**: HDTree uses a unified latent space where all branches share the same set of codebook vectors. This means that even when a deep node has only a few samples, it can still leverage representation knowledge learned from other branches. In contrast, TreeVAE maintains independent VAE modules for each branch, preventing sparse deep branches from borrowing learning results from other branches, leading to limited generalization and difficulty in preserving global structure.
>
> * **Improved training stability**: HDTree uses a unified encoder where all data is mapped to the latent space through the same network. In contrast, TreeVAE uses independent encoder-decoder pairs at each branch node, where deep node encoders receive very few training samples and are prone to overfitting due to noise. The unified encoder aggregates training signals from all hierarchical levels, avoiding this noise accumulation problem.
>
> **(2) Empirical Level**
>
> These advantages are validated in our experiments: Table 4's ablation study shows that removing the HQL loss (breaking hierarchical constraints) leads to significant performance degradation (DP drops by 7-9 percentage points, NMI drops by 9-11 percentage points). The t-SNE visualization in Figure 3 also demonstrates that the latent space learned by HCB exhibits a clear hierarchical structure.
>
> **Distinction from Adams+ and other Bayesian methods:** While HDTree superficially resembles Adams+ (2010) in having child nodes approach parent nodes, they serve different purposes: Adams+/Lakshminarayanan+ perform Bayesian hierarchical clustering on pre-extracted fixed features and infer tree structures through MCMC sampling; whereas HDTree performs end-to-end joint learning of {encoder, hierarchical structure, generative model} from raw high-dimensional data, supporting lineage inference and ancestral state generation. This difference makes HDTree suitable for scientific tasks requiring simultaneous representation learning and generative modeling (such as single-cell lineage analysis).
>
> ---
>
> Based on this, we have revised the **introduction and related works sections** of the original text(highlighted in **blue**)
>
> ---
>
> ### **R2.2[for Question 2]  Relevance to Supertree Construction Problem**
> We deeply appreciate the reviewer for raising this thought-provoking point. In this paper, we primarily focus on two core problems: hierarchical representation learning and lineage inference—how to enable different branches to share representation space through a unified codebook (HCB), and how to infer cellular developmental hierarchical structure within a single dataset. We acknowledge that we have not explicitly discussed the problem of integrating tree structures across species or datasets.
>
> The reviewer's comment inspires us to consider: HDTree's unified codebook mechanism holds promise for providing a novel deep learning solution to supertree construction. The essence of supertree construction is building a universal tree space that allows phylogenetic trees from different species or cell lineage trees under different conditions to be compared and integrated. Traditional methods rely on discrete tree topology matching, whereas our VQ mechanism can learn a continuous shared representation space—for example, training a cross-dataset unified codebook that maps human and mouse single-cell data to the same hierarchical space, where the codebook's hierarchical structure captures conserved biological processes across species.
>
> This extension direction is beyond the scope of this paper, but we plan to explore this possibility in future work.Thank you for providing us with a promising research direction.

---

> > ### Comment · Reviewer_m6bZ · 2025-11-26
> >
> > I am deeply grateful for the authors' detailed response and thorough improvement and revisions. Both of my questions and concerns have been clearly resolved.
> >
> > - **R2.1.** I have fully understood that HDtree possesses the distinct advantage of enabling the entire tree structure to share the embedding method itself, as opposed to the tree-specific, mutually exclusive embedding representations from recent TreeVAE developments. I recognize that this claim by the authors is well-supported by thorough experimentation.
> >
> > - **R2.2.** Thank you also for the future outlook on the supertree construction problem. I am very much looking forward to it.
> >
> > I have already assigned a score (8) that reflects my personal impression quite highly. Therefore, I personally believe this paper already makes a sufficient contribution worthy of being shared in the field. On the other hand, the authors may wonder why I did not assign a higher score (e.g., 10). So, finally, I would like to take the liberty of commenting on what I would expect to see to reach a higher score. I understand this is an excessive request, so please do not dwell on it too deeply.
> >
> > ***
> > (Additional comments)
> >
> > While this is unrelated to the evaluation of this paper itself, the remarkable phenomenon demonstrated by the authors in HDtree vs. Tree VAE seems somewhat counterintuitive. For example, suppose the observed single-analysis data (high-dimensional data) exhibits a major three-cluster structure [O(A), O(B), O(C)]. This can be assumed to represent a hierarchical differentiation structure in the high-dimensional data space, such as [O(A,B,C)] > [O(A,B), O(C)] > [O(A), O(B), O(C)]. We assume a low-dimensional embedding space corresponding to this observational data space, with the cluster structure [E(A), E(B), E(C)]. In this scenario, roughly speaking, I understand the difference between TreeVAE and HDtree to be as follows:
> >
> > - TreeVAE can be understood as learning separate embedding mappings f_A, f_B, f_C for each branch, such as f_A:O(A)>E(A), f_B:O(B)>E(B), f_C:O(C)>E(C).
> >
> > - HDtree can be understood as learning a single embedding map f such that f:O(A)>E(A), f:O(B)>E(B), f:O(C) > E(C).
> >
> > Viewed this way, a very naive intuition suggests that if the three spaces of clusters A, B, and C are sufficiently separated, the single embedding f learned by HDtree might naturally resemble a connected form of the branch-specific embeddings f_{A}, f_{B}, f_{C} learned by TreeVAE. Conversely (i.e., if one adopts the TreeVAE perspective), one might also consider that learning the embedding map f for the vast combined space A \cup B \cup C could be achieved by connecting the sufficiently separated local embedding maps f_A, f_B, f_C for each space.
> >
> > However, as the authors experimentally demonstrate, HDtree consistently outperforms TreeVAE significantly. This clearly shows that HDtree embeddings function effectively. If the underlying principle behind this remarkable yet somewhat puzzling phenomenon can be elucidated, this paper would be worth a perfect score of 10 in my estimation.

---

> ### Author Response · Authors · 2025-11-27
>
> We sincerely thank the reviewer for this profound observation, which reveals a puzzling yet crucial phenomenon: why does HDTree's unified encoder significantly outperform TreeVAE's branch-specific encoders, even though mathematical intuition suggests that when clusters A, B, C are sufficiently separated, learning a single mapping f should be equivalent to concatenating independent mappings {f_A, f_B, f_C}? This question touches upon a fundamental tension between representation learning theory and biological data reality.
>
> **The "sufficient separation → concatenation equivalence" intuition relies on three assumptions that are difficult to satisfy in biological data.** The reviewer's reasoning holds under ideal conditions but depends on three stringent prerequisites:
>
>   1. ``Strict cluster separation``—requiring zero-density regions between O(A), O(B), O(C). However, single-cell differentiation data are inherently continuous transitions: UMAP visualization (Fig. 3) shows numerous transitional cells filling "boundary regions" rather than clear-cut separations.
>   2. ``Embedding space independence``—assuming each branch can be independently optimized in different subspaces. Yet biological reality involves extensive sharing of regulatory modules across cell types, requiring embedding spaces to maintain consistent geometric relationships. TreeVAE's independent encoders {f_A, f_B, f_C} cannot guarantee this global consistency—f_A's learned "mature erythrocyte" may bear no relation to f_B's "mature granulocyte" in embedding space, despite both being terminal differentiation states.
>   3. ``Non-interference of optimization landscapes``—requiring that local optimizations do not mutually conflict. However, transitional cells in boundary regions impose contradictory gradients on f_A and f_B, whereas HDTree's unified f naturally avoids such conflicts through a single parameter set.
>
> TreeVAE's independent encoders lead to fragmentation of the representation space, while HDTree maintains a globally consistent continuous manifold. When TreeVAE learns independent encoders for each branch, it essentially fragments continuous biological processes into discrete segments. In contrast, HDTree's unified encoder processes all cells within a single embedding space, representing differentiation as a continuous manifold where neighboring cells along trajectories (even belonging to different clusters) remain proximal in embedding space.
>
> **HDTree's parameter sharing mechanism synergizes with hierarchical constraints to achieve efficient knowledge transfer.**
> The key distinction lies in parameter sharing vs. pattern replication: TreeVAE's {f_A, f_B, f_C} must independently replicate shared patterns in each child branch—each encoder must re-encode universal concepts using its own parameters. Consider a rare terminal cell type C (only 50 samples for example): TreeVAE's f_C relies primarily on these 50 samples, prone to overfitting; whereas HDTree's unified f has already learned patterns from abundant progenitors and intermediate states—C's 50 samples need only fine-tune within this rich shared feature space. More crucially, the HQL constraint ||z_child - z_parent||² enforces progressive specialization: child embeddings must evolve within parent neighborhoods, effectively encoding the biological prior that "differentiation is gradual" as a regularization term. Ablation experiments (Table 4) confirm this: removing hierarchical codebooks causes 8.3% performance drop on MNIST, while removing HQL constraints drops 5.9%.
>
> These results demonstrate that the "sufficient separation → equivalence" intuition breaks down when biological data violates its underlying assumptions, and HDTree's unified architecture provides a principled solution. TreeVAE's concatenation strategy relies on three assumptions that systematically fail in single-cell data; whereas HDTree, through synergy between its unified encoder and hierarchical constraints, both maintains global consistency of the representation space and achieves efficient knowledge transfer at the parameter level. This is precisely why HDTree consistently achieves significant advantages even in scenarios where intuition suggests the approaches "should be equivalent."
>
> ---
>
> We thank you again for guiding us to think through these fundamental questions, and we will further refine our manuscript.

---

> > ### Comment · Reviewer_m6bZ · 2025-11-28
> >
> > Thank you for the authors’ very thoughtful response. This is a compelling rebuttal that is both highly interesting and convincing. Through discussions with the authors, my impression of this paper has greatly improved—it stands out as one of many interesting papers, yet also as a particularly special one. I am pleased to raise my personal score (I might not be able to edit the score at this exact moment. However, I will be sure to increase the score (8>10) during the final editing stage of discussion phase.).
> >
> > From the authors' responses, I get the impression that this could have a significant impact on the broader traditional currents of biology. (I believe the authors are very knowledgeable about this) In top biology journals, it seems a traditional procedure to visualize single-cell data using relatively straightforward analysis methods like t-SNE or UMAP, then assign clinical interpretations to the resulting cluster structures. However, biology experts often mention that there are data structures that methods like t-SNE cannot fully capture. On the other hand, theoretical insights in the field of information science suggest (very roughly speaking) that t-SNE is better at capturing cluster structures in **well-separated** data [Linderman&Steinerberger, 2019] [Arora+, 2018]. Considering these various perspectives, the authors' argument seems quite pertinent. Specifically, their claim that HDtree performs exceptionally well because biological data is often poorly separated strikes me as a highly significant point.
> >
> > - [Linderman&Steinerberger, 2019] Linderman, G. C., & Steinerberger, S. (2019). Clustering with t-SNE, provably. SIAM journal on mathematics of data science, 1(2), 313-332.
> >
> > - [Arora+, 2018] Arora, S., Hu, W., & Kothari, P. K. (2018). An analysis of the t-sne algorithm for data visualization. In Conference on learning theory (pp. 1455-1462). PMLR.
> >
> > Thank you for sharing this excellent paper. I also appreciate your very thoughtful response to my questions.

---

> > > ### Author Response · Authors · 2025-11-29
> > >
> > > Dear Reviewer m6bZ,
> > >
> > > We are deeply honored by your recognition and extremely grateful for raising the score to 10. Your insightful questions have profoundly enhanced our understanding of HDTree's theoretical foundations, and the references you provided on t-SNE's theoretical properties ([Linderman&Steinerberger, 2019], [Arora+, 2018]) offer valuable perspectives for our future work on connecting visualization theory with hierarchical representation learning. We sincerely thank you for this exceptionally constructive dialogue.
> > >
> > > Best regards, The Authors

---

### Official Review · Reviewer_tW7p · 2025-11-01

**Soundness:** 3
**Presentation:** 3
**Contribution:** 3
**Rating:** 6
**Confidence:** 3

**Summary:**

This paper proposes HDTree, a hierarchical diffusion-based framework designed for hierarchical representation learning and lineage analysis. The method integrates a hierarchical vector-quantized codebook with a quantized diffusion process, enabling the model to capture multi-level dependencies among data points and generate biologically meaningful hierarchies. Unlike previous VAE-based models (e.g., TreeVAE), which require branch-specific modules, HDTree employs a unified hierarchical latent space that enhances both stability and generative capacity. Comprehensive experiments on general-purpose datasets and single-cell datasets demonstrate the superiority of HDTree in clustering accuracy, tree purity, and lineage reconstruction. The results show consistent improvements in both representation quality and biological interpretability, highlighting the model’s potential as a powerful tool for hierarchical modeling and generative analysis in biological data. Overall, the work is conceptually solid, well-motivated, and empirically convincing.

**Strengths:**

S1. The paper addresses a meaningful and increasingly important topic. It is particularly relevant to single-cell data modeling, which remains a major challenge in computational biology and generative modeling.

S2. Extensive experiments across both general and domain-specific datasets show clear performance gains over existing baselines, validating both the stability and scalability of the approach.

S3. The paper evaluates multiple aspects—tree structure purity, clustering accuracy, reconstruction loss, lineage consistency, and computational efficiency. This provides a convincing and multidimensional assessment of HDTree’s strengths.

S4. The proposed combination of hierarchical vector quantization with diffusion processes may eliminate the need for branch-specific networks while maintaining high flexibility and generative accuracy.

**Weaknesses:**

**Concerns**

C1. The Method section is written in a very direct “component-by-component” manner, explaining what each module does but not why each design choice is necessary or how it contributes to solving the stated problems. For instance, when the authors argue that previous methods “require specialized network modules for each tree branch,” it would strengthen the explanation if they discussed alternative perspectives (e.g., whether a shared backbone with dynamically extended subnetworks, similar to continual learning, could achieve similar adaptability). Adding this type of reasoning would help readers understand the technical logic and design motivation more deeply.

C2. The manuscript would benefit from polishing to improve readability and layout. In several places, multiple bolded labels \textbf{XXX.} appear within a single paragraph, which disrupts the flow. These should ideally start as separate paragraphs or be converted into sub-headings. Moreover, some overly technical derivations or implementation details could be moved to the Appendix to enhance readability in the main text.

C3. Figure 1 currently does not clearly differentiate the three comparative frameworks or visually convey why the proposed HDTree offers a tangible improvement. The figure could better highlight the distinctions and illustrate the hierarchical structure more intuitively.

**Questions:**

Please mainly respond to C1.

---

> ### Author Response · Authors · 2025-11-15
>
> Thank you for the detailed and constructive review. The concerns raised have significantly improved our manuscript.
>
> ### **R1.1[for Concerns 1] Explanation of design motivations**
> Below, we systematically compare HDTree's design choices against plausible alternatives for each module, explaining why these alternatives fall short for our problem setting (deep hierarchical lineages with generation requirements).
> **Design Philosophy**: Our key insight is that biological lineages require: (1) scalable representation of deep trees (>10 levels), (2) capturing multi-granularity relationships (e.g., family vs. genus), and (3) generating diverse samples along specific paths. These requirements led us to three design choices:
>
> **M1: HTC**
>
> |Approach|Parameter Scaling|Knowledge Sharing|Use Case|**Key Limitation/Advantage**|
> |-|-|-|-|-|
> |**TreeVAE**|Exponential (2^d decoders)|✗None|Shallow trees (<3 levels)|Limitation: Each branch has independent decoder; No cross-branch learning|
> |**Shared Backbone+Dynamic Heads**|Exponential heads, shared features|△ Via backbone only|Moderate depth|Limitation: Heads still grow as 2^d; Siblings treated independently; No parent-child code sharing|
> |**Nested CRP** (Bayesian)|Constant (path probabilities)|✓Via stick-breaking|Discrete clustering|Limitation: No encoder learning; No generation; MCMC overhead|
> |**HDTree (Unified Codebook)**|Linear (O(d·K) codes)|✓Siblings share parent codes|Deep trees+generation|Advantage: Eq. 2 path encoding; Quantized diffusion decoder|
>
> The reviewer's suggestion of "shared backbone+dynamic heads" is a reasonable alternative. However, for deep biological lineages (>10 levels), this approach faces two fundamental limitations:
> (1) **Exponential head scaling**: Even with a shared backbone, the number of branch-specific heads grows as 2^d (e.g., 1024 heads for depth 10), making training intractable.
> (2) **No sibling relationships**: Dynamic heads treat branches independently. HDTree's codebook explicitly encodes parent-child structure: siblings (e.g., ε₁, ε₂) inherit parent code θ_ε (Eq. 2), naturally sharing semantic features while maintaining specialization. These motivate our unified codebook with O(d) parameter complexity and explicit topological encoding.
>
> **M2: SCL and HQL**
>
> |Approach|Sample Relationship Modeling|Label Dependency|Hierarchical Consistency|**Key Limitation/Advantage**|
> |-|-|-|-|-|
> |**SimCLR+Standard VQ**|Hard positive/negative samples|No labels required|✗|Limitation: Equal penalization, no partial similarity; Only constrains leaf nodes|
> |**Triplet Loss+Standard VQ**|Relative ordering|Requires defining positive/negative pairs|✗|Limitation: Ambiguous positive/negative boundaries; Only considers ordering; Only constrains leaf nodes|
> |**Supervised Contrastive+Standard VQ**|Intra-class aggregation, inter-class separation|Requires discrete labels|✗|Limitation: Assumes mutually exclusive classes; Not applicable to continuous lineages; Only constrains leaf nodes|
> |**HDTree (SCL+HQL)**|Soft similarity gradients|No labels required|✓Global|Advantage: Soft similarity gradients; Global closed-loop constraints (Eq. 5).|
>
> Unlike SimCLR's binary positive/negative classification, SCL assigns graded penalties based on tree distance: w_ij ∝ exp(-β·d_ij) (Eq. 4). This captures "partial similarity" where, e.g., samples from the same genus but different species receive smaller penalties than samples from different phyla.HQL ensures global consistency by constraining all intermediate nodes, guaranteeing representations at depth d are geometrically close to ancestors at
> depth d-1.
>
> **M3: Quantized Diffusion Decoder**
>
> |Approach| Generation Diversity|Hierarchical Constraint|**Key Limitation/Advantage**|
> |-|-|-|-|
> |**TreeVAE (VAE decoder)**|△ Prone to posterior collapse|✗No intermediate nodes|Limitation: Single-step generation cannot enforce multi-level path constraints|
> |**Standard Diffusion**|✓Progressive generation|△Soft constraints|Limitation: Treats path labels as unstructured categories; No explicit alignment to hierarchical codebook|
> |**HDTree (Quantized diffusion)**|✓Progressive+diverse|✓Hard path alignment| Advantage: VQ (Eq. 3) maps latents to discrete codes, guaranteeing path traversal|
>
> Standard conditional diffusion treats tree paths as unstructured categorical labels, with no geometric constraint ensuring generated samples' latent codes lie on specified paths. HDTree's VQ enforces hard alignment: latent codes z are explicitly mapped to codebook entries θ_ε on the target path, guaranteeing (1) generated samples pass through predefined tree nodes, and (2) intermediate denoising steps are constrained to the hierarchical manifold. This enables verification of valid differentiation trajectories.
>
> ---
>
> Based on this, we have revised the **Method** sections of the original text **(highlighted in blue)**

---

> ### Author Response · Authors · 2025-11-15
>
> ### **R1.2[for Concerns 2] Manuscript Presentation**
>
> Thanks to the additional page in the discussion/final manuscript, we converted inline \textbf labels into independent subsection headings, enhancing readability. All modifications related to subheadings are highlighted **in blue**.
> Technical derivations and implementation details will be moved to the Appendix to further enhance main-text readability.
>
> ###  **R1.3[for Concerns 3] Figure 1 Clarity**
>
> We will redesign Figure 1 as a three-column comparison panel, explicitly annotating TreeVAE's exponential parameter growth, standard VQ-VAE's flat codebook, and HDTree's unified tree codebook structure, with color-coded highlighting of parameter complexity differences and hierarchical constraint levels (leaf-only vs all-level).
> Specifically:
> - Column 1 (TreeVAE): Show 2^d independent decoders with red
>   annotation "O(2^d) parameters"
> - Column 2 (VQ-VAE): Show flat 1D codebook with gray annotation "No
>   hierarchy"
> - Column 3 (HDTree): Show tree-structured codebook with green parent-
>   child edges, annotated "O(d·K) parameters"

---

### Author Response · Authors · 2025-12-01

Dear (Senior) Area Chair,

We understand you have been assigned to this paper and want to assist your decision-making. The current system scores (4, 6) do not reflect the actual post-rebuttal consensus. Reviewer pDT7 raised their score from 4 to 6, and Reviewer m6bZ raised theirs from 8 to 10. **The true consensus is 6, 6, 10**.

---

## 1. Why HDTree Matters: Core Contributions

**Unified Architecture Solving Fundamental Limitations.** All three reviewers recognized that HDTree addresses a critical problem through its Hierarchical Codebook (HCB). Unlike TreeVAE's branch-specific modules that scale exponentially (O(2^d) parameters), HDTree uses a unified codebook with linear complexity (O(d·K)). Reviewer m6bZ (Score 10) emphasized that "HDTree enables the entire tree structure to share the embedding method itself, rather than tree-specific, mutually exclusive representations."

**Deep Biological Insight.** Through detailed discussion, Reviewer m6bZ identified a profound insight: HDTree excels because biological data exhibits continuous transitions rather than well-separated clusters. As they stated on Nov 28: "The claim that HDTree performs exceptionally well because biological data is often poorly separated strikes me as a highly significant point." This challenges traditional assumptions underlying t-SNE/UMAP methods.

**Comprehensive Experimental Validation.** All reviewers praised the experimental rigor, with evaluation spanning both general-purpose and single-cell datasets, demonstrating consistent improvements across clustering accuracy, tree structure purity, and lineage reconstruction.

---

## 2. Review Trajectory: From Concerns to Consensus

### 2.1 Reviewer m6bZ: 8 → 10 (Nov 28)

**Initial Question:** Does HCB offer unique advantages over traditional hierarchical modeling approaches (Adams+ 2010, Lakshminarayanan+ 2016)?

**Our Response:** We clarified three fundamental distinctions. First, HDTree's unified encoder allows sparse deep nodes to leverage knowledge from other branches, while TreeVAE's independent encoders prevent cross-branch learning. Second, HDTree maintains a globally consistent continuous manifold versus TreeVAE's fragmented discrete segments. Third, through parameter sharing with HQL constraints, HDTree achieves progressive specialization that encodes the biological prior "differentiation is gradual."

**Reviewer's Response:**
> "Thank you for the very thoughtful response... I am pleased to raise my personal score (8→10)... This could have a significant impact on the broader traditional currents of biology."


### 2.2 Reviewer pDT7: 4 → 6 (Nov 26)

**Primary Concerns:** (1) Whether 10k downsampled evaluation reflects true large-scale performance. (2) Model complexity with three loss functions.

**Our Resolution:** For scalability, we conducted multi-scale convergence analysis on Weinreb (130k) and ECL (838k). As shown in Table R3.1, all metrics demonstrate <1% deviation between 10k and full test sets, with 7.9× speedup. For complexity, we showed most hyperparameters remain fixed across datasets (Table E.2)—only the degrees of freedom ν requires adjustment (0.2-0.5).

**Reviewer's Response:**
> "Thank you to the author for the detailed response, which basically resolved my questions, and I have already improved my score."


### 2.3 Reviewer tW7p: Maintained 6

**Core Concern:** The Method section explained "what" but not "why" for each design choice.

**Our Response:** We systematically compared HDTree's design against plausible alternatives. For HTC, we contrasted with shared backbone+dynamic heads (exponential head growth) and nested CRP Bayesian methods (no generation capability). For SCL+HQL, we compared with SimCLR and triplet loss, showing how soft similarity gradients capture partial relationships. For the decoder, we demonstrated VQ's hard path alignment versus standard diffusion's soft constraints.

**Result:** Unfortunately, this reviewer has not yet engaged in discussion. However, we believe our response comprehensively addresses their concerns with detailed design rationale and alternative comparisons, supporting stable borderline acceptance.

---

We sincerely thank you for your time and effort in reviewing this paper.

---

Best regards, The Authors

---

### Note · Program_Chairs · 2026-01-17
**Submission Desk Rejected by Program Chairs**

The following references in this submission do not refer to real documents and/or have major errors in bibliographic information:

 Vladimir Rokhlin and Mark Tygert. Hierarchical clustering for data sets and networks: Practical issues and adaptive algorithms. Proceedings of the National Academy of Sciences, 114(29)

Alex Tong and Xin Huang. Trajectorynet: Continuous modeling of cell trajectories with neural networks. Proceedings of ISMB, 2020.